# DMT-JEPA: Learning Discriminative Masked Targets for Joint-Embedding Predictive Architecture

## Abstract

The joint-embedding predictive architecture (JEPA) recently has shown impressive results in extracting visual representations from unlabeled imagery under a masking strategy. However, we reveal its disadvantages, notably its insufficient understanding of local semantics. This deficiency originates from masked modeling in the embedding space, resulting in a reduction of discriminative power and can even lead to the neglect of critical local semantics. To bridge this gap, we introduce DMT-JEPA, a novel masked modeling objective rooted in JEPA, specifically designed to generate discriminative latent targets from neighboring information. Our key idea is simple: we consider a set of semantically similar neighboring patches as a target of a masked patch. To be specific, the proposed DMT-JEPA (a) computes feature similarities between each masked patch and its corresponding neighboring patches to select patches having semantically meaningful relations, and (b) employs lightweight cross-attention heads to aggregate features of neighboring patches as the masked targets. Consequently, DMT-JEPA highlights that increased discriminative power of target representations benefits a diverse spectrum of downstream tasks. Through extensive experiments, we demonstrate our effectiveness across various visual benchmarks, including ImageNet-1K image classification, ADE20K semantic segmentation, and COCO object detection tasks. Code is available at: https://anonymous.4open.science/r/DMT-JEPA-anony.

## 1 Introduction

The success of self-supervised learning (SSL) frameworks (Chen et al., 2020; Chen & He, 2021; He et al., 2020; Grill et al., 2020), especially in harnessing vast reservoirs of unlabeled images, has been undeniable in the computer vision community. Model architectures like the Vision Transformer (ViT; (Dosovitskiy et al., 2021)) have consistently garnered significant attention, and initial attempts at seamless integration with SSL have indeed demonstrated potential (Chen et al., 2021; Xie et al., 2021; Caron et al., 2021). In particular, Masked autoencoder (MAE; (He et al., 2021)), which reconstructs missing patches on pixel space, has achieved advanced success in various visual downstream tasks, such as image classification, object detection, and semantic segmentation.

Unlike classic Masked Autoencoders (MAE) that reconstruct raw pixels, forcing the model to dedicate significant capacity to high-frequency, low-level visual details, I-JEPA (Assran et al., 2023) operates entirely in the embedding space. This latent predictive approach encourages the network to discard pixel-level noise and directly learn abstract, high-level semantics. However, while I-JEPA successfully avoids pixel redundancy, we demonstrate that its independent patch-level targets still lack the local cohesiveness required for optimal dense prediction, motivating our discriminative target approach. Recently, the image-based joint-embedding predictive architecture (I-JEPA) has shown promising results in learning self-supervised representations by leveraging a masking strategy to reconstruct representations of masked patches. Nevertheless, we observed that this approach often results in performances that are less compatible during fine-tuning, especially when compared to the pixel reconstruction method (*e.g.*, MAE), and also results in indistinct attention maps, as illustrated in Figure 2. These indistinctnesses can be attributed to insufficient discriminative power, which is crucial for a deep understanding of local semantics.

One challenge posed by existing SSL approaches built upon ViTs is the potential lack of local semantics in the extracted representations from disjoint input patches, where local semantics are naturally intertwined within the image patches. This inspiration leads us to integrate explicit processing of local semantics, aiming to produce discriminative latent targets that enhance masked modeling for predicting the embedding of masked images from unmasked ones. To avoid the creation of non-discriminate latent targets, our principal strategy involves generating a semantically meaningful target for each masked patch. This strategy, inspired by Yun et al. (2022), utilizes the similarities among patches within a specific neighborhood to capture local semantics comprehensively. During pre-training, we leverage these self-supervised latent targets to capture local semantics, thereby offering benefits for various tasks, including dense prediction and image classification.

In this paper, we introduce the Discriminative Masked Targets for Joint-Embedding Predictive Architecture (DMT-JEPA), a novel self-supervised representation learning framework focused on latent reconstruction for a masked image. Our goal is to generate discriminative latent targets that can capture local semantics that can serve as an alternative masked modeling objective, which can be incorporated with the joint-embedding predictive architecture (LeCun, 2022). To this end, we propose Masked Semantic Neighboring to find semantically similar neighboring patches for masked patches and Local Aggregation Target to generate the discriminative dense targets from them. Specifically, Masked Semantic Neighboring computes feature similarities between each masked patch and its corresponding neighboring patches to select semantically similar patches, and Local Aggregation Target employs lightweight cross-attention heads to aggregate features of chosen neighboring patches as the latent targets for masked patches. Consequently, the proposed DMT-JEPA not only exhibits superior efficiency comparable to I-JEPA (Assran et al., 2023) but also achieves enhanced efficacy in learning self-supervised representations, highlighting that increased discriminability of targets leads to improved performance, as illustrated in Figure 2. This significant improvement is advantageous across a broad spectrum of tasks, encompassing image classification as well as dense prediction downstream tasks such as semantic segmentation and object detection.

To demonstrate the effectiveness of DMT-JEPA, we conduct extensive downstream experiments after pre-training ViT-B/16, ViT-L/16, and ViT-H/14 on ImageNet-1k; these experiments include ImageNet-1K image classification, COCO object detection, ADE20K semantic segmentation, DAVIS video segmentation, and Clevr local prediction benchmarks. Our experimental results demonstrate that the DMT-JEPA improves the performance of I-JEPA with a large margin and even outperforms other SSL baselines (He et al., 2021; Chen et al., 2021; Bao et al., 2021) on various benchmarks; for example, our method achieved +1.4 mIoU (*i.e.,* $47.6 \rightarrow 49.0$) on ADE20K semantic segmentation, +1.7 $(\mathcal{J}\&\mathcal{F})_m$ (*i.e.,* $56.6 \rightarrow 58.3$) on DAVIS video segmentation, +1.0 $\text{AP}^{\texttt{box}}$ (*i.e.,* $49.9 \rightarrow 50.9$) on COCO object detection, and +1.1 $\text{AP}^{\texttt{mask}}$ (*i.e.,* $44.5 \rightarrow 45.6$) on COCO instance segmentation. In addition, we observed that further increasing the discriminative power of targets leads to additional performance gains, as detailed in Table D.5. Furthermore, we observed that DMT-JEPA can even outperform MAE in ImageNet fine-tuning experiments, achieving scores of 84.6 (on ViT-B/16) and 86.6 (on ViT-L/16), in contrast to MAE's scores of 83.6 (on ViT-B/16) and 85.9 (on ViT-L/16). This demonstrates that the proposed method not only benefits dense representation learning but also enhances the quality of global image representations. In summary, this paper contributes the following:

- We propose DMT-JEPA, a novel self-supervised learning framework that enhances the Joint-Embedding Predictive Architecture by constructing discriminative latent targets.

- We introduce two core modules: *Masked Semantic Neighboring*, which identifies semantically consistent patches within a local neighborhood, and *Local Aggregation Target*, which aggregates these patches via a lightweight cross-attention mechanism.

- We demonstrate through extensive experiments that DMT-JEPA significantly outperforms strong baselines, including I-JEPA and MAE. Our method achieves superior results across diverse downstream tasks, validating its efficacy for both dense prediction and global classification.

## 2 Related Work

**Self-supervised Visual Representation Learning.** Early self-supervised approaches primarily relied on contrastive learning (Chen et al., 2020; He et al., 2020; Chen et al., 2021) or clustering (Caron et al., 2020) to

align representations of augmented views. To capture local-to-global correspondence, DINO (Caron et al., 2021) utilized a momentum encoder with multi-crop training to achieve knowledge distillation in Vision Transformers (ViT). Building on this, DINOv2 (Oquab et al., 2023) recently demonstrated that scaling such discriminative self-supervised approaches with extensive curated data can yield all-purpose features. Meanwhile, methods like iBOT (Zhou et al., 2022) attempted to bridge the gap between masked modeling and distillation by conducting masked prediction on tokenizer outputs. SelfPatch (Yun et al., 2022) further explored enforcing invariance between a patch and its neighbors to enhance semantic consistency. While we draw inspiration from leveraging neighboring information, DMT-JEPA fundamentally differs from SelfPatch: we focus on predicting information about neighboring patches within a predictive architecture, without relying on a global contrastive loss. This distinct approach enhances computational efficiency, enabling significant improvements in dense prediction tasks where global-only objectives often fall short.

**Masked Image Modeling.** Following the success of BERT in NLP, MIM has become a dominant paradigm in vision (Bao et al., 2021; He et al., 2021; Xie et al., 2022). The seminal MAE (He et al., 2021) and SimMIM (Xie et al., 2022) demonstrated that simple pixel-level reconstruction of masked patches promotes the learning of transferable representations. BEiT (Bao et al., 2021) and subsequent works (Peng et al., 2022) proposed reconstructing discrete visual tokens derived from a dVAE or a semantic tokenizer. However, pixel-based reconstruction often focuses on high-frequency details rather than semantic abstraction. To address this, recent works have shifted toward predicting latent features. Data2vec (Baevski et al., 2022) proposes predicting the representations of the momentum encoder for masked tokens, unifying modalities under a single framework. Similarly, MaskFeat (Wei et al., 2022) predicts HOG features to reduce the focus on raw pixel redundancy. Despite these advances, these methods typically operate as autoencoders. Our work aligns with the shift toward latent targets but integrates them into a joint-embedding predictive framework to better capture local semantic structures.

**Joint-Embedding Predictive Architectures.** Proposed by LeCun (2022), JEPA aims to learn representations by predicting the embeddings of a signal from a compatible context without explicitly reconstructing the input signal. I-JEPA (Assran et al., 2023) instantiated this for images, predicting the representations of masked blocks using a context encoder. This approach avoids the computational overhead of pixel decoders and the semantic gap of pixel reconstruction. Recently, V-JEPA (Bardes et al., 2024b) extended this philosophy to video, learning temporal feature prediction solely from non-semantic localized objectives. However, a limitation of current JEPA implementations is that the target representations, being outputs of a target encoder, may lack sufficient discriminability or local semantic cohesiveness, potentially leading to indistinct attention maps. Unlike I-JEPA, which predicts independent block representations, or MC-JEPA (Bardes et al., 2024a), which focuses on motion-content separation, our DMT-JEPA explicitly constructs *discriminative dense targets* by aggregating semantically similar neighbors. This allows the model to learn more robust local semantics for efficient predictive architectures and dense recognition tasks.

## 3 Method

In this section, we present a novel masked modeling framework, coined DMT-JEPA, designed for the joint-embedding predictive architecture to enhance understanding local semantics within images, as shown in Figure 1. Our key idea is that semantically similar representations can provide local semantics as a masked modeling objective by enforcing them to have similar representations. We first provide preliminaries in Section 3.1 and then present details of two modules, Masked Semantic Neighboring in Section 3.2 and Local Aggregation Target in Section 3.3.

### 3.1 Preliminaries

We first describe the problem setup and notations and then revisit the Image-based Joint-Embedding Predictive Architectures (I-JEPA; (Assran et al., 2023)), which is a self-supervised visual representation learning under masked modeling.

**Problem Setup and Notations.** Given an image with a dimension of $3 \times H \times W$ and a patch resolution of $P$, our goal is to learn a masked autoencoder framework with an encoder $f_e(\cdot)$ and a decoder $f_d(\cdot)$ to

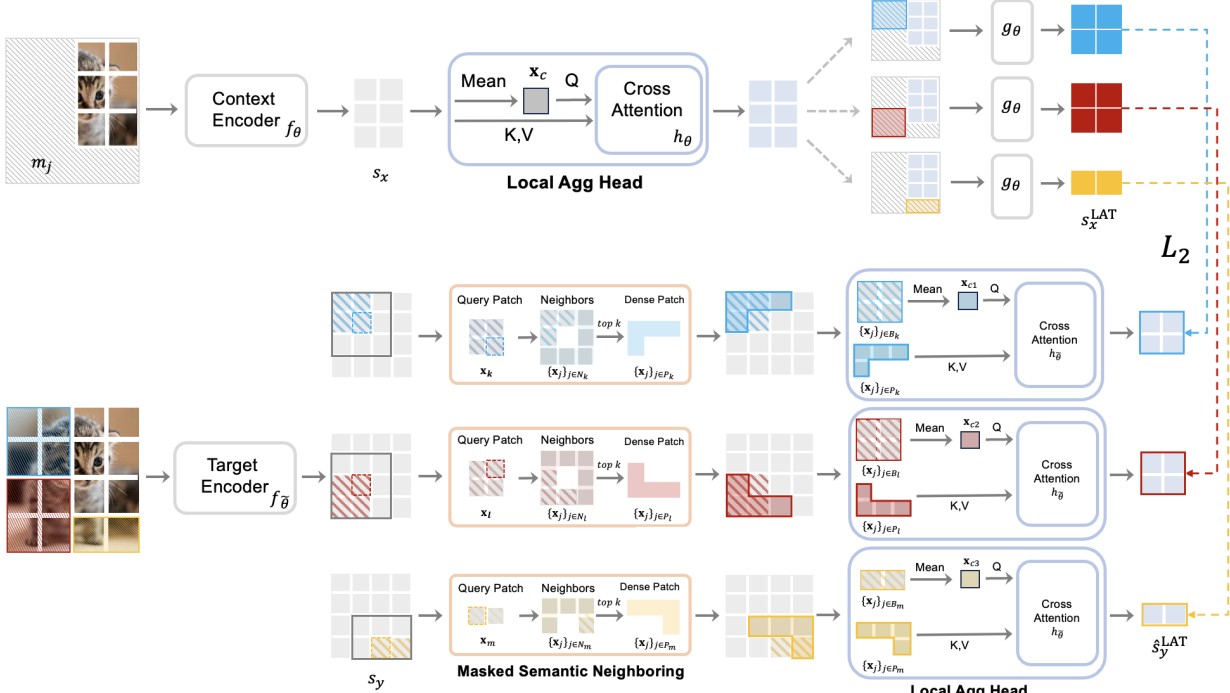

Figure 1: **Illustration of the proposed novel masked modeling framework (DMT-JEPA), rooted in I-JEPA, designed for discriminative latent targets from neighboring information.** The Masked Semantic Neighboring module computes the dense semantic similarity between the query patch and its neighboring patch based on representations from the target encoder $f_{\tilde{\theta}}(\cdot)$ to select semantically similar patches from the neighborhood. Then Local Aggregation Target module composed of a context patch aggregation head $h_{\theta}(\cdot)$ and a target patch aggregation head $h_{\tilde{\theta}}(\cdot)$, aggregates target features of selected patches using cross-attention to construct dense targets. Finally, the model is optimized by the average $L_2$ distance between the predicted dense representations and the target dense representation.

recover the masked patches using unmasked ones. We formally denote patch embeddings of raw input via each linear projection layer, *i.e.*, $\mathbf{x} \in \mathbb{R}^{N \times D}$, $H$ and $W$ are the height and width of each image, and $D$ is the dimension of features. Note that $N = H/P \times W/P$ and $N$ is the total number of patches.

**Masked Autoencoder.** To address the masked image modeling problem, MAE (He et al., 2021) first applied a random masking set $M$ along the total number of patches, and then an encoder to extract features from unmasked patches. Finally, unmasked embeddings and masked tokens were concatenated into a decoder to recover the raw pixels of masked patches. The vanilla masking loss for each image is calculated with the mean square loss between the targeted $\mathbf{p}_i$ and predicted normalized pixels $\hat{\mathbf{p}}_i$ as:

$$\mathcal{L}_{\texttt{MAE}} = \frac{1}{|M|} \sum_{i \in M} ||\mathbf{p}_i - \hat{\mathbf{p}}_i||_2^2, \tag{1}$$

where $|M|$ denotes the total number of masked patches in the masking set $M$.

**Image-based Joint-Embedding Predictive Architecture.** To tackle the masked image modeling task, I-JEPA (Assran et al., 2023) introduced a context encoder $f_{\theta}(\cdot)$, a target encoder $f_{\tilde{\theta}}(\cdot)$, and a predictor $g_{\theta}(\cdot)$, to predict the $M$ target block representations $\mathbf{s}_y(1), ..., \mathbf{s}_y(M)$ given the output of the context encoder, $\mathbf{s}_x$. For a target block $\mathbf{s}_{y_i}$ corresponding to a target mask $\mathcal{B}_i$, the predictor $g_{\theta}(\cdot, \cdot)$ takes as input the output of the context encoder $\mathbf{s}_x$ and a mask token for each patch to predict $\{\mathbf{m}_j\}_{j \in \mathcal{B}_i}$, and outputs the patch-level prediction $\{\hat{\mathbf{s}}_{y_j}\}_{j \in \mathcal{B}_i}$, that is, $\{\hat{\mathbf{s}}_{y_j}\}_{j \in \mathcal{B}_i} = g_{\theta}(\mathbf{s}_x, \{\mathbf{m}_j\}_{j \in \mathcal{B}_i})$. The masking objective is optimized by the average $L_2$ distance between the predicted patch-level representations $\hat{\mathbf{s}}_{y_j}$ and the target patch-level

representation $\mathbf{s}_{y_j}$, which is formulated as:

$$\mathcal{L}_{\texttt{I-JEPA}} = \frac{1}{|M|} \sum_{i=1}^{M} \sum_{j \in \mathcal{B}_i} ||\mathbf{s}_{y_j} - \hat{\mathbf{s}}_{y_j}||_2^2, \tag{2}$$

where $|M|$ denotes the total number of target blocks, and $\mathcal{B}_i$ is the mask corresponding to the $i$-th target block.

## 3.2 Masked Semantic Neighboring

However, a masked modeling target in the representation space like I-JEPA could pose a challenge in terms of missing local semantics if the target patch-level representations $\mathbf{s}_{y_j}$ were less discriminative among themselves. As shown in Figure 2, we also observed that I-JEPA often generates indistinct attention maps, and it arguably indicates its deficiency in comprehending local semantics. To tackle this, we aim to generate target representations capturing local semantics that can serve as an alternative masked modeling objective, which can be incorporated with the joint-embedding predictive architecture (LeCun, 2022). We note that prior investigation on patch-level representation learning (Yun et al., 2022) inspires us to explore similarities among patch-level representations located in a neighborhood. To this end, we propose Masked Semantic Neighboring module to find semantically similar neighboring patches for masked patches and Local Aggregation Target module (see Section 3.3) to make them have similar target representations.

**Masked Semantic Neighboring.** For patches in a given masked block, we aim to find their neighboring patches semantically similar, as neighboring patches often share a semantic context. In order to sample semantically similar patches from the neighborhood $\mathcal{N}_i$, we compute the dense semantic similarity $d(i,j)$ between the query patch $\mathbf{x}_i$ and its neighboring patch $\mathbf{x}_j$ for all $j \in \mathcal{N}_i$ based on representations from the target encoder $f_{\tilde{\theta}}(\cdot)$, which is formulated as:

$$d(i,j) = \frac{f_{\tilde{\theta}}(\mathbf{x}_i)^{\top} f_{\tilde{\theta}}(\mathbf{x}_j)}{\|f_{\tilde{\theta}}(\mathbf{x}_i)\|_2 \|f_{\tilde{\theta}}(\mathbf{x}_j)\|_2}, \tag{3}$$

where $f_{\tilde{\theta}}(\mathbf{x}_i), f_{\tilde{\theta}}(\mathbf{x}_j) \in \mathbb{R}^{1 \times D}$, and $\| \cdot \|_2$ denotes the $\ell_2$-norm operator. With the computed similarity scores, we apply a ranking on the neighboring patches $\{\mathbf{x}_j\}_{j \in \mathcal{N}_i}$ and select a set of dense patches $\{\mathbf{x}_j\}_{j \in \mathcal{P}_i}$ with top-$k$ highest similarities, where $\mathcal{P}_i$ denotes a set of dense patch indices in the neighborhood, and $k$ is the number of dense patches, $i.e.$, $k = |\mathcal{P}_i|$. Unless stated otherwise, we use $k = 4$ for our experiments.

## 3.3 Local Aggregation Target

We remark that our goal is to generate target representations capturing local semantics and discriminative among themselves. With the benefit of the selected neighboring patches having similar semantics, we introduce Local Aggregation Target module composed of a context patch aggregation head $h_\theta(\cdot)$ and a target patch aggregation head $h_{\tilde{\theta}}(\cdot)$. Specifically, we aggregate target representations of selected patches $\{f_{\tilde{\theta}}(\mathbf{x}_j)\}_{j \in \mathcal{P}_i}$ using cross-attention head $h_\theta(\cdot)$ to construct dense target $\mathbf{s}_i^{\texttt{LAT}}$ for enforcing semantically similar patches could have the similar dense targets. Simultaneously, we symmetrically apply context aggregation head $h_\theta(\cdot)$ to produce corresponding context $\mathbf{s}_x^{\texttt{LAT}}$ from context representations of unmasked patches as:

$$\mathbf{s}_i^{\texttt{LAT}} = h_{\tilde{\theta}}(\{\mathbf{x}_j\}_{j \in \mathcal{P}_i}, \mathbf{x}_{c_i}), \quad \mathbf{s}_x^{\texttt{LAT}} = h_\theta(\mathbf{s}_x, \mathbf{x}_c),$$

where $\mathbf{s}_x$ denotes context embeddings and $\mathbf{x}_{c_i}, \mathbf{x}_c$ denote the averaged embeddings from all patches in the target encoder and only unmasked patches in the context encoder, respectively. The cross-attention operator $h_\theta(\cdot)$ and $h_{\tilde{\theta}}(\cdot)$ is formulated as:

$$h_{\tilde{\theta}}(\{\mathbf{x}_j\}_{j \in \mathcal{P}_i}, \mathbf{x}_{c_i}) = \texttt{Softmax}\left(\frac{\mathbf{x}_{c_i} \{\mathbf{x}_j\}_{j \in \mathcal{P}_i}^{\top}}{\sqrt{D}}\right) \{\mathbf{x}_j\}_{j \in \mathcal{P}_i},$$

$$h_\theta(\mathbf{s}_x, \mathbf{x}_c) = \texttt{Softmax}\left(\frac{\mathbf{x}_c \mathbf{s}_x^{\top}}{\sqrt{D}}\right) \mathbf{s}_x, \tag{4}$$

where $D$ is the dimension of embeddings. For a given target block $\mathbf{s}_{y_i}^{\texttt{LAT}}$ corresponding to a target mask $\mathcal{B}_i$, the predictor $g_\theta(\cdot, \cdot)$ takes as input the output of the context patch aggregation head $\mathbf{s}_x^a$ and a mask token for each patch to predict $\{\mathbf{m}_j\}_{j \in \mathcal{B}_i}$, and outputs a dense prediction $\{\hat{\mathbf{s}}_{y_j}^{\texttt{LAT}}\}_{j \in \mathcal{B}_i} = g_\theta(\mathbf{s}_x^{\texttt{LAT}}, \{\mathbf{m}_j\}_{j \in \mathcal{B}_i})$. The new masking objective is optimized by the average $L_2$ distance between the predicted dense representations $\hat{\mathbf{s}}_{y_j}^{\texttt{LAT}}$ and the target dense representation $\mathbf{s}_i$ in the $i$-th block, which is formulated as:

$$\mathcal{L}_{\text{DMT-JEPA}} = \frac{1}{|M|} \sum_{i=1}^{M} \sum_{j \in \mathcal{B}_i} ||\mathbf{s}_i^{\texttt{LAT}} - \hat{\mathbf{s}}_{y_j}^{\texttt{LAT}}||_2^2, \tag{5}$$

where $|M|$ denotes the total number of target blocks, and $\mathcal{B}_i$ is the target mask corresponding to the $i$-th target block. In terms of semantic similarity among patches, the closer the final target representations, pre-training through these targets would promote the enhancement of learned embeddings that encompass local semantics.

### 3.4 Theoretical Analysis

In this subsection, we provide a theoretical justification for the proposed DMT-JEPA framework. We posit that the standard patch-level objective in masked modeling often suffers from high variance due to local visual noise (*e.g.*, texture details, occlusion) that is irrelevant to high-level semantics. We demonstrate that our Local Aggregation Target (LAT) module acts as a semantic denoising operator, stabilizing the learning objective via variance reduction.

**Setup and Assumptions.** Let $\mathcal{Z} \subset \mathbb{R}^D$ be the underlying semantic manifold of the image data. We assume that for any patch $\mathbf{x}_i$, its latent representation $\mathbf{h}_i = f_{\bar{\theta}}(\mathbf{x}_i)$ can be decomposed into a true semantic component $\boldsymbol{\mu}_i \in \mathcal{Z}$ and a noise component $\boldsymbol{\epsilon}_i$, such that:

$$\mathbf{h}_i = \boldsymbol{\mu}_i + \boldsymbol{\epsilon}_i, \tag{6}$$

where $\boldsymbol{\epsilon}_i$ represents high-frequency variations or aleatoric uncertainty unrelated to the broader context. We assume $\boldsymbol{\epsilon}_i$ is a zero-mean random variable with variance $\sigma^2 \mathbf{I}$, and the noise between distinct patches is independent.

**Proposition 1 (Semantic Consistency of Neighbors).** *Let $\mathcal{N}_i$ be the spatial neighborhood of patch $i$. The Masked Semantic Neighboring (MSN) module selects a subset of indices $\mathcal{P}_i \subset \mathcal{N}_i$ based on high cosine similarity. We assume that for selected neighbors $j \in \mathcal{P}_i$, the semantic divergence from the query patch is bounded by a small constant $\delta$, such that $\|\boldsymbol{\mu}_i - \boldsymbol{\mu}_j\|_2 \leq \delta$.*

This proposition formalizes the role of the MSN module: by filtering for high similarity, we ensure that the aggregation set consists of patches that share the same underlying semantic intent $\boldsymbol{\mu}_i$, up to a small divergence $\delta$.

**Theorem 1 (Variance Reduction via Local Aggregation).** Consider the Local Aggregation Target $\mathbf{s}_i^{\texttt{LAT}}$ computed as a weighted sum of $k = |\mathcal{P}_i|$ neighboring representations. Assuming uniform attention weights ($\alpha_j = 1/k$) and allowing for a positive spatial correlation $\rho \in [0, 1)$ among the high-frequency noise of adjacent patches, the expected squared error of the aggregated target with respect to the true semantic component $\boldsymbol{\mu}_i$ is strictly less than that of a single patch target.

*Proof.* Let the single target be $\mathbf{h}_i = \boldsymbol{\mu}_i + \boldsymbol{\epsilon}_i$. The expected error is $\mathbb{E}[\|\mathbf{h}_i - \boldsymbol{\mu}_i\|^2] = \mathbb{E}[\|\boldsymbol{\epsilon}_i\|^2] = D\sigma^2$. Now consider the aggregated target $\mathbf{s}_i^{\texttt{LAT}} \approx \frac{1}{k} \sum_{j \in \mathcal{P}_i} (\boldsymbol{\mu}_j + \boldsymbol{\epsilon}_j)$. Using the bias-variance decomposition:

$$\mathcal{E}_{\text{LAT}} = \left\| \frac{1}{k} \sum_{j \in \mathcal{P}_i} (\boldsymbol{\mu}_j - \boldsymbol{\mu}_i) \right\|^2 + \text{Var}\left( \frac{1}{k} \sum_{j \in \mathcal{P}_i} \boldsymbol{\epsilon}_j \right)$$

Unlike perfectly independent noise, spatially adjacent patches share correlated textures. Assuming an average positive spatial correlation coefficient $\rho$ among the noise terms in $\mathcal{P}_i$, the variance expands to:

$$\text{Variance} = \frac{1}{k^2} \sum_{j \in \mathcal{P}_i} \mathbb{E}[\|\boldsymbol{\epsilon}_j\|^2] + \frac{1}{k^2} \sum_{p \neq q \in \mathcal{P}_i} \text{Cov}(\boldsymbol{\epsilon}_p, \boldsymbol{\epsilon}_q) = D\sigma^2 \left( \frac{1}{k} + \frac{k-1}{k}\rho \right).$$

Thus, the total error is bounded by:

$$\mathcal{E}_{\text{LAT}} \leq \delta^2 + D\sigma^2 \left( \frac{1}{k} + \frac{k-1}{k}\rho \right).$$

Because $\rho < 1$, the variance multiplier $\left( \frac{1}{k} + \frac{k-1}{k}\rho \right)$ is strictly less than 1. Provided the semantic divergence $\delta$ is sufficiently small relative to this variance gap, $\mathcal{E}_{\text{LAT}} < D\sigma^2$, confirming that LAT provides a closer approximation to the true semantic signal. ∎

**Theorem 2 (Implicit Smoothness Regularization).** *Minimizing the DMT-JEPA objective $\mathcal{L}_{\text{DMT-JEPA}}$ encourages the predictor $g_\theta$ to learn a smoother mapping with respect to the spatial lattice, acting as an empirical regularizer against high-frequency visual noise.*

*Proof Sketch.* Let $\mathbf{s}_i^{\text{LAT}}$ and $\mathbf{s}_j^{\text{LAT}}$ be the targets for two spatially adjacent patches $i$ and $j$. Because these targets are derived from a localized consensus, they share a significant overlap in their semantic neighborhoods ($\mathcal{P}_i \approx \mathcal{P}_j$). Consequently, the distance between the aggregated targets $\|\mathbf{s}_i^{\text{LAT}} - \mathbf{s}_j^{\text{LAT}}\|^2$ is bounded and smoothly varying, parameterized by the non-overlapping fraction $\gamma_{ij}$. Since the objective explicitly minimizes $\|\hat{\mathbf{s}}_{y_j}^{\text{LAT}} - \mathbf{s}_i^{\text{LAT}}\|^2$ over the data distribution, the predictor $g_\theta$ is implicitly regularized to track these smooth target transitions. This prevents the model from overfitting to abrupt, high-frequency fluctuations caused by individual noisy patches, encouraging local smoothness across the context-to-target mapping. ∎

## 4 Experiments

**Evaluation Roadmap:** We sequentially evaluate the framework on dense prediction tasks (segmentation and detection), followed by global representation evaluations (classification), low-level geometry tasks, and conclude with comprehensive module ablations.

### 4.1 Experimental setup

**Datasets.** Following previous methods (He et al., 2021; Assran et al., 2023), we use ImageNet-1K (Deng et al., 2009) for image classification, MS-COCO (Lin et al., 2014) for object detection and instance segmentation, and ADE20K (Zhou et al., 2017; 2018) for semantic segmentation. We closely follow previous work (Chen et al., 2021; Xie et al., 2021; Caron et al., 2021), and adopt the Mask R-CNN (He et al., 2017) as the detector. The ViT-Base (Dosovitskiy et al., 2021) backbone weights are initialized with weights pre-trained on ImageNet-1K using our DMT-JEPA. Following the settings in (He et al., 2021; Bao et al., 2021), we use the UPerNet approach (Xiao et al., 2018) based on our ImageNet-1K pre-trained ViT-Base for evaluation. For a fair comparison, we fine-tune the detector with the same learning rate in (He et al., 2021; Bao et al., 2021). For video object segmentation, we use DAVIS-2017 (Pont-Tuset et al., 2017) dataset containing 60 training, 30 validation, and 60 testing videos. For local prediction tasks on Clevr (Johnson et al., 2016), we follow the previous work (Assran et al., 2023) and use Clevr/Count and Clevr/Dist.

**Evaluation Metrics.** We follow previous masked image modeling work (He et al., 2021; Bao et al., 2021) to report the classification accuracy of linear probing and fine-tuning. For object detection and instance segmentation on MS-COCO, we apply $\text{AP}^{\texttt{box}}$ and $\text{AP}^{\texttt{mask}}$ as metrics for the bounding boxes and the instance masks. mIoU results are reported to evaluate semantic segmentation on ADE20K. For video object segmentation on DAVIS-2017, we use Jabri-based $(\mathcal{J}\&\mathcal{F})_m$, $\mathcal{J}_m$, $\mathcal{F}_m$ as metrics to evaluate the quality of frozen representations of image patches by segmenting scenes with the nearest neighbor between consecutive frames. For local prediction tasks on Clevr, we use object counting and depth prediction to evaluate the linear probing performance of our model.

Table 1: **ADE20K semantic segmentation, COCO object detection, and instance segmentation.** We fine-tuned pre-trained ViT-B/16 models to perform ADE20K semantic segmentation and COCO object detection and instance segmentation. The mIoU, $AP^{box}$, and $AP^{mask}$ metrics denote the results of ADE20K segmentation, COCO detection, and segmentation, respectively. The best are indicated in **bold**.

| Method | Pre-train data | mIoU | $AP^{box}$ | $AP^{mask}$ |
|---|---|---|---|---|
| Supervised | ImageNet-1K w/ labels | 47.4 | 47.9 | 42.9 |
| DINO (Caron et al., 2021) | ImageNet-1K | 46.8 | 50.1 | 43.4 |
| MoCo v3 (Chen et al., 2021) | ImageNet-1K | 47.3 | 47.9 | 42.7 |
| BEiT (Bao et al., 2021) | ImageNet-1K+DALLE | 47.1 | 49.8 | 44.4 |
| MAE (He et al., 2021) | ImageNet-1K | 48.1 | 50.3 | 44.9 |
| I-JEPA (Assran et al., 2023) | ImageNet-1K | 47.6 | 49.9 | 44.5 |
| DMT-JEPA (ours) | ImageNet-1K | **49.0** | **50.9** | **45.6** |

Table 2: **DAVIS video object segmentation.** We perform DAVIS 2017 video object segmentation using ImageNet-1K pre-trained ViT-B/16, ViT-L/16, and ViT-H/14 odels. We report Jabri-based metrics $(\mathcal{J}\&\mathcal{F})_m$, $\mathcal{J}_m$, $\mathcal{F}_m$ to evaluate the quality of pre-trained representations. The best results are indicated in **bold**.

| Method | Backbone | $(\mathcal{J}\&\mathcal{F})_m$ | $\mathcal{J}_m$ | $\mathcal{F}_m$ |
|---|---|---|---|---|
| MAE (He et al., 2021) | ViT-B/16 | 51.0 | 49.4 | 52.6 |
| | ViT-L/16 | 53.4 | 52.5 | 54.3 |
| I-JEPA (Assran et al., 2023) | ViT-B/16 | 56.2 | 56.1 | 56.3 |
| | ViT-L/16 | 56.6 | 56.3 | 56.9 |
| | ViT-H/14 | 57.8 | 57.7 | 57.9 |
| DMT-JEPA (ours) | ViT-B/16 | **57.7** | **56.7** | **58.7** |
| | ViT-L/16 | **58.3** | **57.3** | **59.2** |
| | ViT-H/14 | **59.8** | **59.5** | **60.1** |

**Implementation.** For input images, the resolution is resized to $224 \times 224$, *i.e.*, $H = W = 224$. We follow prior work (He et al., 2021; Assran et al., 2023) and apply a patch size of 16, *i.e.*, $P = 16$. The small, base, and large models of ViT (Dosovitskiy et al., 2021) architecture are used for experiments. We set the embedding dimension of the predictor to 384, and keep the number of self-attention heads the same as the backbone context-encoder. For the smaller ViT-S/16 and ViT-B/16 context-encoder, we set the depth of the predictor as 6. For ViT-L/16 context-encoders, we set the depth of the predictor to 12. Following I-JEPA (Assran et al., 2023), we use AdamW to optimize the context-encoder and predictor weights. We train our model using the default batch size of 2048, and the learning rate linearly increased from 1e-4 to 1e-3 during the first 15 epochs of pre-training, and decay to 1e-6 following a cosine schedule. The weight decay is linearly increased from 0.04 to 0.4, and the target-encoder weights are initialized the same as the context-encoder weights, and updated via an exponential moving average. We use a momentum value of 0.996, and linearly increase this value to 1.0. For masking, we use the same strategy and settings as I-JEPA (Assran et al., 2023) for 4 possibly overlapping target blocks masks. Our small, base, and large models are pre-trained on ImageNet-1K (Deng et al., 2009) for 600 epochs.

## 4.2 Comparison to prior work

In this work, we propose a novel and effective discriminative dense target for latent reconstruction-based masked modeling with a joint-embedding predictive architecture. In order to demonstrate the effectiveness of the proposed DMT-JEPA, we comprehensively compare it to previous mask image modeling baselines (He et al., 2021; Baevski et al., 2022; Chen et al., 2022; Assran et al., 2023).

**Detection and Segmentation tasks.** For the ADE20K semantic segmentation and COCO object detection & instance segmentation benchmarks, we report the quantitative comparison results in Table 1; our method achieved the best results regarding all the metrics compared to previous mask modeling baselines. In particular, the proposed DMT-JEPA outperforms I-JEPA (Assran et al., 2023), the current image-based joint-embedding predictive architecture by 0.9@mIoU. Also, we achieve significant performance gains of

Table 3: **ImageNet-1K image linear classification.** We perform a linear evaluation on pre-trained ViT-B/16 and ViT-L/16 models for image classification on ImageNet-1K benchmark. We report the top-1 accuracy to evaluate the quality of pre-trained representations. The best results are indicated in **bold**.

| Method | Backbone | Epochs | Top-1 Acc |
|---|---|---|---|
| data2vec (Baevski et al., 2022) | ViT-L/16 | 1600 | 77.3 |
| MAE (He et al., 2021) | ViT-B/16 | 1600 | 68.0 |
| | ViT-L/16 | 1600 | 76.0 |
| I-JEPA (Assran et al., 2023) | ViT-B/16 | 600 | 72.9 |
| | ViT-L/16 | 600 | 77.5 |
| | ViT-H/14 | 300 | 79.3 |
| DMT-JEPA (ours) | ViT-B/16 | 600 | **73.8** |
| | ViT-L/16 | 600 | **78.2** |
| | ViT-H/14 | 300 | **80.6** |

Table 4: **ImageNet-1K image fine-tuning classification.** We perform fine-tuning pre-trained ViT-B/16 and ViT-L/16 models for image classification on ImageNet-1K benchmark. We report the top-1 accuracy to evaluate the quality of fine-tuned representations. The best results are indicated in **bold**.

| Method | Pre-train data | Backbone | Top-1 Accuracy |
|---|---|---|---|
| DINO (Caron et al., 2021) | ImageNet-1K | ViT-B/16 | 82.8 |
| MAE (He et al., 2021) | ImageNet-1K | ViT-B/16 | 83.6 |
| DMT-JEPA (ours) | ImageNet-1K | ViT-B/16 | **84.6** |
| MAE (He et al., 2021) | ImageNet-1K | ViT-L/16 | 85.9 |
| DMT-JEPA (ours) | ImageNet-1K | ViT-L/16 | **86.6** |
| MAE (He et al., 2021) | ImageNet-1K | ViT-H/14 | 86.9 |
| DMT-JEPA (ours) | ImageNet-1K | ViT-H/14 | **87.9** |

$1.0@\text{AP}^{\text{box}}$ and $1.1@\text{AP}^{\text{mask}}$ on COCO object detection and instance segmentation compared to I-JEPA, which indicates the importance of leveraging the (self-supervised) discriminative dense targets to capture local semantics for dense prediction tasks. Furthermore, we observed that DMT-JEPA even can achieve better results than the strongest baseline, MAE (He et al., 2021), a generative autoencoder architecture for masked image modeling.

In addition, our method shows significant and consistent gains in the DAVIS 2017 video object segmentation benchmark as shown in Table 2. Compared to I-JEPA, ours achieved the results gains of $1.5@(\mathcal{J}\&\mathcal{F})_m$, $0.6@\mathcal{J}_m$, and $2.4@\mathcal{F}_m$ on ViT-B/16. Moreover, the margins increased more significantly when we evaluated the large-scale backbone, ViT-L/16, by $1.7@(\mathcal{J}\&\mathcal{F})_m$, and ViT-H/14, by $2.1@(\mathcal{J}\&\mathcal{F})_m$, which shows a scaling behavior of ours. Overall, these significant improvements reported in Tables 1 and 2 highlight the superiority of our approach in capturing local semantics during self-supervised pre-training. This stands in sharp contrast to I-JEPA, which performs even worse than MAE in the same evaluations.

**ImageNet linear classification task.** Here, we validate the quality of our learned global representation by performing the common linear evaluation task on ImageNet-1k. Table 3 summarizes the results; DMT-JEPA outperforms all the baselines in Table 3. For example, DMT-JEPA achieved 78.2% top-1 accuracy on ViT-L/16, while MAE and I-JEPA did 76.0% and 77.5%, respectively. These results further indicate the benefit of the proposed method in learning the global semantics of images.

**ImageNet fine-tuning comparisons.** For a comprehensive comparison with DINO and MAE, we follow previous works (Caron et al., 2021; He et al., 2021) and fine-tune pre-trained ViT-B/16 and ViT-L/16 on ImageNet-1K. Table 4 reports the comparison results with prior approaches using DINO and MAE pre-trained weights. For instance, our DMT-JEPA outperforms other models with the highest scores of 84.6% on ViT-B/16 and 86.6% on ViT-L/16, surpassing DINO and MAE, which achieved scores of 82.8% and 83.6% on the ViT-B/16 model, respectively. These significant improvements underscore our framework's superior capability in grasping the global semantics of images, in contrast to I-JEPA, which has reported finetuning results that are less effective than those achieved by MAE.

Table 5: **Clever object counting and depth prediction.** We perform a linear evaluation on pre-trained models for Clever object counting and depth prediction benchmarks. The Clevr/Count and Clevr/Dist metrics denote the result of object counting and depth prediction tasks, respectively. The best results are indicated in **bold**, and the second best ones are underlined.

| Method | Backbone | Clevr/Count | Clevr/Dist |
|---|---|---|---|
| DINO (Caron et al., 2021) | ViT-B/8 | 86.6 | 53.4 |
| iBOT (Zhou et al., 2022) | ViT-L/16 | 85.7 | 62.8 |
| data2vec (Baevski et al., 2022) | ViT-L/16 | 85.3 | 71.3 |
| MAE (He et al., 2021) | ViT-B/16 | **86.6** | **70.8** |
| | ViT-L/16 | **92.1** | **73.0** |
| I-JEPA (Assran et al., 2023) | ViT-B/16 | 82.2 | 70.7 |
| | ViT-L/16 | 85.6 | 71.2 |
| | ViT-H/14 | 86.7 | 72.4 |
| DMT-JEPA (ours) | ViT-B/16 | 83.5 | 71.1 |
| | ViT-L/16 | 87.1 | 71.8 |
| | ViT-H/14 | **87.9** | **73.1** |

Table 6: **Analysis on computational costs.** We perform computational analyses on pre-trained ViT-B/16 models for comparison with MAE and I-JEPA. The best results are indicated in **bold**.

| Method | Pre-train Epochs | Total Steps | Max Memory per GPU (GB) | Running Time per Step (ms) |
|---|---|---|---|---|
| MAE (He et al., 2021) | 1600 | 500,800 | **11.5** | **235.2** |
| I-JEPA (Assran et al., 2023) | **600** | **375,600** | 21.9 | 606.4 |
| DMT-JEPA (ours) | **600** | **375,600** | 21.9 | 608.2 |

**Other low-level tasks.** We also present additional Clevr benchmarks for measuring abilities of object-counting and depth prediction. Table 5 shows linear evaluation results of DMT-JEPA on the Clevr counting and depth benchmarks. Compared to I-JEPA (Assran et al., 2023), we achieve the results gains of 1.3@Clevr/Count and 0.4@Clevr/Dist using ViT-B/16. Moreover, we observe similar scaling behavior in Table 2, with increased improvements on ViT-L/16, while our model on ViT-H can outperform MAE. These results show that JEPA objectives on the embedding space have the potential to degrade dense representation learning and outperform view-invariance compared to pixel reconstruction.

**Computational Comparison with MAE & I-JEPA.** To comprehensively assess the efficiency of the proposed DMT-JEPA, we compare it against both MAE (He et al., 2021) and I-JEPA (Assran et al., 2023) across total pre-training epochs, total optimization steps, max memory per GPU, and running time per step in Table 6. While MAE remains the fastest per step due to its asymmetric encoder processing only unmasked patches, DMT-JEPA compensates by requiring significantly fewer total optimization steps to reach convergence (375,600 vs. 500,800). Furthermore, compared to I-JEPA, the current state-of-the-art image-based joint-embedding predictive architecture, our method introduces negligible overhead, maintaining virtually identical memory footprint and wall-clock time. More importantly, for this comparable computational cost, DMT-JEPA unlocks significantly richer representations, yielding superior downstream performance in segmentation and detection (Tables 1 & 2), image classification (Tables 3 & 4), and local geometry prediction (Table 5). These analyses demonstrate that our discriminative aggregation module is a highly efficient structural tradeoff for enhanced dense prediction capabilities.

For a more comprehensive comparison with DINOv2 (Oquab et al., 2023), a recent strong self-supervised baseline trained on a large-scale dataset, we evaluate our models under both linear evaluation and fine-tuning protocols on ADE20K semantic segmentation. The comparison results are reported in Table 7. Under fair linear evaluation settings using the ViT-L/16 backbone, our DMT-JEPA consistently outperforms DINOv2 across different pre-training resolutions, achieving 45.6 mIoU at 224 resolution (+2.6 mIoU) and 49.7 mIoU at 416 resolution (+3.5 mIoU). Furthermore, under the full fine-tuning protocol at 224 resolution, DMT-JEPA with ViT-L/16 achieves 53.6 mIoU, significantly surpassing the 50.3 mIoU achieved by DINOv2. It

Table 7: **ADE20K semantic segmentation context.** We present linear and fine-tuning DMT-JEPA models alongside DINOv2 (Oquab et al., 2023).The best results in this table are indicated in **bold**.

| Method | Pre-train data | Backbone | Pre-train Res. | Eval Protocol | mIoU |
|---|---|---|---|---|---|
| DINOv2 (Oquab et al., 2023) | ImageNet-1K | ViT-L/16 | 224 | Linear Eval | 43.0 |
| | ImageNet-1K | ViT-L/16 | 416 | Linear Eval | 46.2 |
| | ImageNet-1K | ViT-L/16 | 224 | Fine-tuning | 50.3 |
| DMT-JEPA (ours) | ImageNet-1K | ViT-L/16 | 224 | Linear Eval | 45.6 |
| | ImageNet-1K | ViT-L/16 | 416 | Linear Eval | 49.7 |
| | ImageNet-1K | ViT-B/16 | 224 | Fine-tuning | 49.0 |
| | ImageNet-1K | ViT-L/16 | 224 | Fine-tuning | 53.6 |
| | ImageNet-1K | ViT-H/16 | 224 | Fine-tuning | **58.2** |

Table 8: **Ablation studies on component analysis.** We perform ablation studies on Masked Semantic Neighboring (MSN) and Local Aggregation Target (LAT) modules using a pre-trained ViT-S/16 model on the DAVIS benchmark. The best results are indicated in **bold**.

| MSN | LAT | $(\mathcal{J}\&\mathcal{F})_m$ | $\mathcal{J}_m$ | $\mathcal{F}_m$ |
|---|---|---|---|---|
| ✗ | ✗ | 53.7 | 52.5 | 54.8 |
| ✓ | ✗ | 55.2 | 54.3 | 56.1 |
| ✗ | ✓ | 54.6 | 53.3 | 55.9 |
| ✓ | ✓ | **57.1** | **55.7** | **58.5** |

is also worth noting that even our base model (ViT-B/16) achieves a highly competitive 49.0 mIoU, while scaling up to ViT-H/16 yields the best overall performance at 58.2 mIoU. These results clearly demonstrate that our framework's ability to learn local semantics during self-supervised pre-training provides substantial and consistent gains for downstream dense prediction tasks.

## 4.3 Experimental Analysis

In this section, we performed ablation studies to demonstrate the benefit of the proposed Masked Semantic Neighboring and Local Aggregation Target modules. Here, we conducted extensive experiments on ImageNet-1k pre-trained ViT-S/16 to explore the impact of cross-attention and pooling, types of local aggregation heads, and learned meaningful patch-level representations.

**Masked Semantic Neighboring & Local Aggregation Target.** In order to demonstrate the effectiveness of Masked Semantic Neighboring (MSN) and Local Aggregation Target (LAT), we ablate the impact of each module and report the quantitative results on DAVIS 2017 video object segmentation benchmark with ViT-S/16 in Table 8. As shown in the table, adding MSN to the vanilla baseline highly increases the results of $1.5@(\mathcal{J}\&\mathcal{F})_m$, which validates the benefit of masked semantic neighboring in finding semantically similar neighboring patches for masked patches. Meanwhile, introducing only LAT in the baseline increases the video segmentation performance regarding all metrics. More importantly, incorporating MSN and LAT into the baseline significantly raises the performance by $3.4@(\mathcal{J}\&\mathcal{F})_m$. These improving results validate the importance of MSN and LAT in extracting local semantics from (self-supervised) discriminative dense targets for better representations.

**Visualizations of Learned Attention Maps.** Learning discriminative attention maps is one of the key aspects of capturing local semantics for downstream tasks of dense prediction type, such as segmentation and detection. To better evaluate the quality of learned attention maps, we visualize the averaged maps from different heads in the last attention layer by using a pre-trained ViT-B/16 target encoder. For a more comprehensive comparison, we also add the attention maps from I-JEPA (Assran et al., 2023) target encoder. The qualitative visualization results are shown in Figure 2. Note that columns for each image sample represent the original image, attention maps from the target encoder in I-JEPA, and attention maps from the target encoder in our DMT-JEPA. As can be seen, attention maps from the target encoder in our DMT-JEPA are discriminative and focus on the object semantics in the image. However, the attention maps

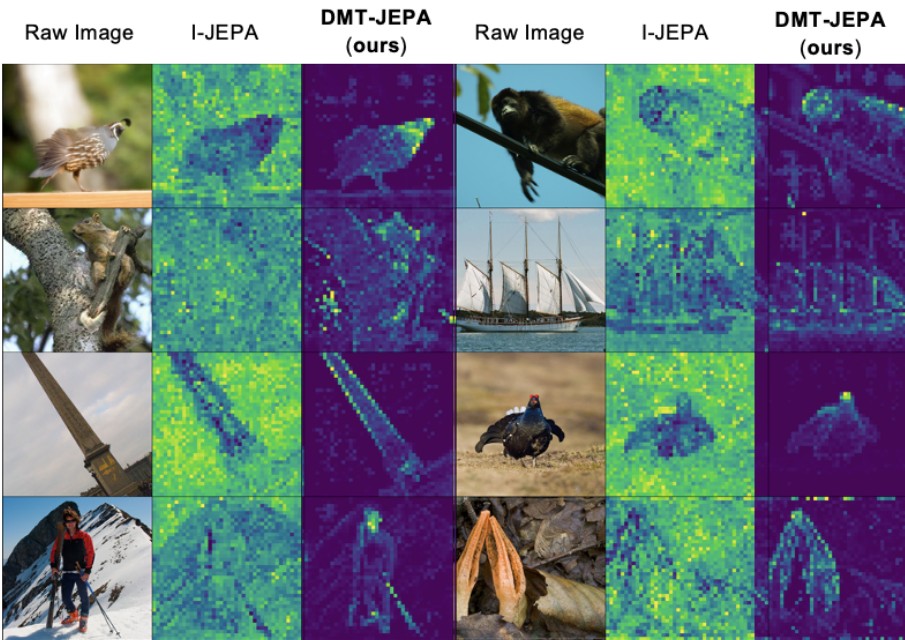

Figure 2: **Qualitative visualization of learned attention maps using ViT-B/16 model.** Columns for each sample denote the original image, attention maps from I-JEPA (Assran et al., 2023), and attention maps from our DMT-JEPA. Our DMT-JEPA achieves much better attention maps.

from the target encoder in I-JEPA activate both the foreground and background and can not effectively discriminate the object semantics, because they did not incorporate local semantics into target representations as our DMT-JEPA did. Meanwhile, the attention maps in our DMT-JEPA have more focus on the details of foreground objects than that from the target encoder, indicating the effectiveness of our local aggregation target module in generating target representations with local semantics. In each example, the attention maps from both the target encoder and the local aggregation head of our model sharply focus on key object regions, accurately highlighting fine-grained local semantics. This refined focus is enabled by aggregating contextually similar neighboring patches within our Local Aggregation Target module, significantly enhancing the semantic coherence and discriminative quality of the resulting target representations. Such highly discriminative representations not only confirm the effectiveness of our aggregation strategy in capturing precise local semantics but also explain why our DMT-JEPA consistently achieves enhanced performance on downstream tasks.

## 5  Conclusion

We introduce a novel discriminative embedding objective tailored for masked modeling on the joint-embedding predictive architecture from unlabeled images. To tackle this, we aim to produce semantically meaningful target representations in a self-supervised manner by leveraging the prior that neighboring patches often contain similar semantics. To be specific, we first search semantically similar patches for a masked patch within its neighborhood by computing their similarities on the representation space. Then we generate the aggregated representations from the selected neighboring patches to serve as a masked modeling objective via a lightweight cross-attention head. Finally, the proposed objective would accelerate learned representations of semantically similar patches being closer, which can be advantageous in understanding local semantics within images. Through extensive experiments, we have demonstrated our models are not only effective in dense prediction types of downstream tasks but also show strong performance in image classification tasks. We believe that our work would highlight the effectiveness of considering a discriminative target for masked modeling on the embedding space.

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

## Appendix

In this appendix, we provide additional implementation and dataset details in Section A, the complete algorithm for DMT-JEPA in Section B, theoretical discussions in Section C, extended experimental analyses in Section D, and qualitative visualization analyses in Section E, and more discussions in Section F.

## A  Implementation and Dataset Details

### A.1  Pre-training Settings

We utilize the Vision Transformer (ViT) architecture (Dosovitskiy et al., 2021) as our backbone.

The implementation is built upon the PyTorch framework.

**Data Augmentation.** During pre-training, we apply standard self-supervised learning augmentations. The input images are randomly cropped with a scale range of $[0.2, 1.0]$ and resized to $224 \times 224$. We apply random horizontal flipping with a probability of 0.5. Color jittering is applied with a strength of 0.4 for brightness, contrast, and saturation, and 0.1 for hue. We also utilize random grayscale conversion with a probability of 0.2 and solarization with a probability of 0.0. Unlike contrastive methods (*e.g.*, MoCo, SimCLR), we do not apply Gaussian blur, following the I-JEPA protocol.

**Optimization.** We use the AdamW optimizer with a batch size of 2048 distributed across GPUs. The learning rate follows a linear warmup for the first 15 epochs, reaching a peak base learning rate of $1.0 \times 10^{-3}$ (scaled linearly with batch size: $lr = base\_lr \times batch\_size/256$), and then decays to $1.0 \times 10^{-6}$ via a cosine schedule. The weight decay increases linearly from 0.04 to 0.4 over the course of training. The momentum parameter for the target encoder update starts at $\tau_0 = 0.996$ and linearly increases to 1.0.

## A.2  Downstream Task Details

**ImageNet-1K Linear Probing.** To evaluate the quality of the frozen representations, we train a linear classifier on top of the global average pooled features from the last layer of the frozen backbone. We train for 100 epochs using the SGD optimizer with a batch size of 16,384. The learning rate is initialized to 0.1 and decays following a cosine schedule. We apply no weight decay and use only random resized cropping and flipping for data augmentation, consistent with standard protocols (He et al., 2021; Assran et al., 2023).

**ImageNet-1K Fine-tuning.** For end-to-end fine-tuning, we initialize the model with pre-trained weights and update all parameters. We train for 100 epochs with a batch size of 1,024. We use the AdamW optimizer with a base learning rate of $5 \times 10^{-4}$ (scaled as $lr \times \text{batch\_size}/256$), a weight decay of 0.05, and a layer-wise learning rate decay of 0.65 (for ViT-B) or 0.75 (for ViT-L). We employ strong data augmentation including Mixup ($\alpha = 0.8$), CutMix ($\alpha = 1.0$), RandAugment, and Random Erasing to prevent overfitting, following the recipe in MAE (He et al., 2021).

**COCO Object Detection & Instance Segmentation.** We utilize the Mask R-CNN (He et al., 2017) framework implemented in Detectron2, equipped with a Feature Pyramid Network (FPN). We adapt the ViT backbone using simple feature pyramids as described in (Li et al., 2021). The model is fine-tuned on the COCO `train2017` set for 100 epochs. We use the AdamW optimizer with an initial learning rate of $1 \times 10^{-4}$ and a weight decay of 0.1. We employ large-scale jittering (LSJ) with a scale range of $[0.1, 2.0]$ and crop size of $1024 \times 1024$. During testing, we resize the short side of images to 800 pixels.

**ADE20K Semantic Segmentation.** We use the UPerNet (Xiao et al., 2018) framework for semantic segmentation. We train the model for 160k iterations with a batch size of 16 on the ADE20K `train` set. We use the AdamW optimizer with a learning rate of $3 \times 10^{-4}$ (with polynomial decay power 0.9), a weight decay of 0.05, and a layer-wise learning rate decay of 0.65. The input resolution is set to $512 \times 512$. During inference, we report mIoU using single-scale testing.

**DAVIS-2017 Video Object Segmentation.** Following (Jabri et al., 2020), we evaluate the spatiotemporal correspondence of frozen features without any fine-tuning. We treat video segmentation as a label propagation task. For a query pixel in the current frame, we find the $k$-nearest neighbors ($k = 5$) in the memory bank of past frames and transfer their labels via soft-voting. The frames are resized to a resolution of 480p. We report the region similarity ($\mathcal{J}$) and contour accuracy ($\mathcal{F}$) averaged over the validation set.

**Clevr Local Prediction Tasks.** To assess the model's ability to capture local geometry and counts, we use the Clevr/Count and Clevr/Dist datasets. We freeze the backbone and train a linear regression head on top of the patch-level features. For object counting, we formulate this as a classification task to predict the number of objects in the scene. For depth prediction, we predict the distance of the object center from the camera for every object in the scene. For both tasks, we train for 50 epochs using the Adam optimizer with a learning rate of $1 \times 10^{-3}$.

# B  Algorithm for DMT-JEPA

Algorithm B.1 outlines the complete pre-training procedure for DMT-JEPA.

# C  Theoretical Discussions

## C.1  Expanded Variance Reduction Proof

We expand on the proof of Theorem 1 regarding variance reduction. Recall that our target representation for a patch $i$ is modeled as $\mathbf{h}_i = \boldsymbol{\mu}_i + \boldsymbol{\epsilon}_i$, where $\boldsymbol{\mu}_i$ is the clean semantic signal and $\boldsymbol{\epsilon}_i \sim \mathcal{N}(0, \sigma^2 \mathbf{I})$ is independent noise.

The Mean Squared Error (MSE) of using a single patch $j$ as the target for patch $i$ (where $j$ is a neighbor) is:

$$\text{MSE}_{\text{single}} = \mathbb{E}[\|(\boldsymbol{\mu}_j + \boldsymbol{\epsilon}_j) - \boldsymbol{\mu}_i\|^2] = \|\boldsymbol{\mu}_j - \boldsymbol{\mu}_i\|^2 + \mathbb{E}[\|\boldsymbol{\epsilon}_j\|^2] = \Delta_{ij}^2 + D\sigma^2, \tag{7}$$

---

**Algorithm B.1** DMT-JEPA Pre-training Algorithm

---

**Require:** Unlabeled dataset $\mathcal{D}$, Context Encoder $f_\theta$, Target Encoder $f_{\tilde{\theta}}$, Predictor $g_\theta$, Aggregation Heads $h_\theta, h_{\tilde{\theta}}$.

1: Initialize $\theta$, $\tilde{\theta} \leftarrow \theta$.
2: **for** each batch $\mathbf{x}$ in $\mathcal{D}$ **do**
3:     $\mathbf{x} \leftarrow \text{Augment}(\mathbf{x})$
4:     Create target masks $M$ and context masks $M_c$.
5:     *// Target Encoding*
6:     $\mathbf{s}_y \leftarrow f_{\tilde{\theta}}(\mathbf{x})$ {Full target embeddings}
7:     *// Masked Semantic Neighboring (MSN)*
8:     **for** each target block $i \in M$ **do**
9:         Identify spatial neighborhood $\mathcal{N}_i$.
10:        Compute similarity $d(i,j)$ for $j \in \mathcal{N}_i$.
11:        Select top-$k$ neighbors $\mathcal{P}_i$.
12:     **end for**
13:     *// Local Aggregation Target (LAT)*
14:     $\mathbf{s}_y^{\texttt{LAT}} \leftarrow h_{\tilde{\theta}}(\{\mathbf{s}_{y_j}\}_{j \in \mathcal{P}_i})$ {Generate discriminative targets}
15:     *// Context Encoding & Prediction*
16:     $\mathbf{s}_x \leftarrow f_\theta(\mathbf{x} \odot M_c)$ {Context embeddings}
17:     $\mathbf{s}_x^{\texttt{LAT}} \leftarrow h_\theta(\mathbf{s}_x)$ {Symmetric context aggregation}
18:     $\hat{\mathbf{s}}_y^{\texttt{LAT}} \leftarrow g_\theta(\mathbf{s}_x^{\texttt{LAT}}, \text{MaskTokens})$
19:     *// Optimization*
20:     $\mathcal{L} \leftarrow \sum_i \|\mathbf{s}_{y_i}^{\texttt{LAT}} - \hat{\mathbf{s}}_{y_i}^{\texttt{LAT}}\|_2^2$
21:     Update $\theta \leftarrow \text{Optimizer}(\theta, \nabla_\theta \mathcal{L})$
22:     Update $\tilde{\theta} \leftarrow \tau\tilde{\theta} + (1-\tau)\theta$ {EMA update}
23: **end for**

---

where $\Delta_{ij} = \|\boldsymbol{\mu}_j - \boldsymbol{\mu}_i\|$ is the semantic shift.

For the aggregated target $\mathbf{s}_i^{\texttt{LAT}} = \frac{1}{k}\sum_{j \in \mathcal{P}_i}(\boldsymbol{\mu}_j + \boldsymbol{\epsilon}_j)$, the error is:

$$\text{MSE}_{\text{LAT}} = \mathbb{E}\left[\left\|\frac{1}{k}\sum_{j \in \mathcal{P}_i}\boldsymbol{\mu}_j - \boldsymbol{\mu}_i + \frac{1}{k}\sum_{j \in \mathcal{P}_i}\boldsymbol{\epsilon}_j\right\|^2\right]. \tag{8}$$

Assuming noise independence between patches, the cross-terms vanish:

$$\text{MSE}_{\text{LAT}} = \underbrace{\left\|\frac{1}{k}\sum_{j \in \mathcal{P}_i}(\boldsymbol{\mu}_j - \boldsymbol{\mu}_i)\right\|^2}_{\text{Semantic Bias}} + \underbrace{\frac{1}{k^2}\sum_{j \in \mathcal{P}_i}\mathbb{E}[\|\boldsymbol{\epsilon}_j\|^2]}_{\text{Noise Variance}}. \tag{9}$$

The noise variance term becomes $\frac{1}{k^2} \cdot k \cdot D\sigma^2 = \frac{D\sigma^2}{k}$. The semantic bias term is bounded by $\max_{j \in \mathcal{P}_i}\|\boldsymbol{\mu}_j - \boldsymbol{\mu}_i\|^2 \leq \delta^2$. Thus, we achieve a reduction in total error if $\delta^2 + \frac{D\sigma^2}{k} < D\sigma^2$, which holds true when the semantic neighborhood is tight ($\delta$ is small) and the noise is non-trivial.

## C.2 Lipschitz Continuity and Predictor Smoothness

In this subsection, we provide a detailed derivation for Theorem 2, demonstrating how the Local Aggregation Target (LAT) imposes a smoothness constraint on the predictor function $g_\theta$.

**Definition 1 (Local Lipschitz Continuity).** A function $f : \mathcal{X} \to \mathcal{Y}$ is locally Lipschitz continuous with constant $L$ if for all $\mathbf{x}_1, \mathbf{x}_2$ in a local neighborhood $\mathcal{N}$, $\|\mathbf{f}(\mathbf{x}_1) - \mathbf{f}(\mathbf{x}_2)\|_2 \leq L\|\mathbf{x}_1 - \mathbf{x}_2\|_2$.

**Theorem 2 (Implicit Smoothness Regularization).** *Minimizing the DMT-JEPA objective $\mathcal{L}_{\text{DMT-JEPA}}$ encourages the predictor $g_\theta$ to learn a smoother mapping with respect to the spatial lattice. By providing targets that vary smoothly across adjacent patches, the objective acts as an empirical regularizer against high-frequency visual noise.*

*Proof Sketch.* Let $\mathbf{s}_i^{\text{LAT}}$ and $\mathbf{s}_j^{\text{LAT}}$ be the targets for two spatially adjacent patches $i$ and $j$. The squared Euclidean distance between these targets is:

$$\|\mathbf{s}_i^{\text{LAT}} - \mathbf{s}_j^{\text{LAT}}\|^2 = \left\| \frac{1}{k} \sum_{p \in \mathcal{P}_i} \mathbf{h}_p - \frac{1}{k} \sum_{q \in \mathcal{P}_j} \mathbf{h}_q \right\|^2. \tag{10}$$

Let $\mathcal{O}_{ij} = \mathcal{P}_i \cap \mathcal{P}_j$ be the overlapping set of neighbors. We can decompose the sums into shared and non-shared components:

$$\mathbf{s}_i^{\text{LAT}} - \mathbf{s}_j^{\text{LAT}} = \frac{1}{k} \left( \sum_{p \in \mathcal{P}_i \setminus \mathcal{O}_{ij}} \mathbf{h}_p - \sum_{q \in \mathcal{P}_j \setminus \mathcal{O}_{ij}} \mathbf{h}_q \right). \tag{11}$$

Assuming bounded representations $\|\mathbf{h}\| \leq C$, the norm is bounded by the number of non-overlapping elements:

$$\|\mathbf{s}_i^{\text{LAT}} - \mathbf{s}_j^{\text{LAT}}\| \leq \frac{1}{k} \cdot 2C \cdot (k - |\mathcal{O}_{ij}|) = 2C\gamma_{ij}. \tag{12}$$

This demonstrates that the distance between the aggregated targets is strictly bounded and smoothly varying, parameterized by the non-overlapping fraction $\gamma_{ij}$. Because the objective minimizes the expected error $\mathbb{E}[\|g_\theta(\mathbf{c}_i) - \mathbf{s}_i^{\text{LAT}}\|^2]$ over the data distribution, the predictor $g_\theta$ is implicitly regularized to track these smooth target transitions.

Unlike single-patch targets, where stochastic noise $\boldsymbol{\epsilon}$ can cause the target distance $\|\mathbf{h}_i - \mathbf{h}_j\|$ to jump arbitrarily (violating smoothness), LAT ensures that the learning signal itself remains smooth. Consequently, this prevents the model from overfitting to abrupt, high-frequency fluctuations, encouraging local smoothness across the context-to-target mapping rather than imposing a strict mathematical Lipschitz bound for all arbitrary inputs. ∎

## C.3  Gradient Variance Reduction

We next analyze the impact of the aggregated target on the optimization landscape, specifically regarding the variance of the stochastic gradients.

**Proposition 2 (Reduced Gradient Variance).** *Let $\mathbf{g}_{single}$ and $\mathbf{g}_{LAT}$ be the stochastic gradient estimators for the predictor weights $\theta$ using single-patch targets and LAT targets, respectively. The variance of the gradient norm is strictly reduced under LAT, i.e., $Var(\|\mathbf{g}_{LAT}\|) < Var(\|\mathbf{g}_{single}\|)$.*

*Proof.* Consider the simplified loss for a single sample $\mathcal{L} = \frac{1}{2}\|g_\theta(\mathbf{x}) - \mathbf{y}\|^2$. The gradient with respect to the predictor output is $\nabla_g \mathcal{L} = (g_\theta(\mathbf{x}) - \mathbf{y})$. The variance of this gradient term is directly proportional to the variance of the target $\mathbf{y}$:

$$\text{Var}[\nabla_g \mathcal{L}] = \text{Var}[g_\theta(\mathbf{x}) - \mathbf{y}] = \text{Var}[\mathbf{y}]. \tag{13}$$

From Section C.1, we established that $\text{Var}[\mathbf{y}_{\text{single}}] = \sigma^2$ and $\text{Var}[\mathbf{y}_{\text{LAT}}] \approx \frac{\sigma^2}{k}$. The gradient with respect to model parameters is $\nabla_\theta \mathcal{L} = \nabla_g \mathcal{L} \cdot \frac{\partial g}{\partial \theta}$. Assuming the Jacobian $\mathbf{J} = \frac{\partial g}{\partial \theta}$ is independent of the target noise $\boldsymbol{\epsilon}$, we have:

$$\text{Cov}(\nabla_\theta \mathcal{L}) = \mathbf{J}^\top \text{Cov}(\nabla_g \mathcal{L})\mathbf{J} = \mathbf{J}^\top (\text{Var}[\mathbf{y}]\mathbf{I})\mathbf{J}. \tag{14}$$

Since $\text{Var}[\mathbf{y}_{\text{LAT}}] < \text{Var}[\mathbf{y}_{\text{single}}]$ for $k > 1$, the trace of the covariance matrix (total variance) is reduced by a factor of $1/k$. This reduction in gradient variance implies more stable SGD updates, allowing for potentially larger learning rates and faster convergence during pre-training. ∎

### C.4   Semantic Consistency of Neighbor Selection

Finally, we provide a probabilistic justification for the Masked Semantic Neighboring (MSN) module, ensuring that it selects semantically relevant patches even in the presence of noise.

**Warm-Start via Positional Embeddings.** A natural question arises regarding the behavior of MSN at the very beginning of training (Epoch 0), when the target encoder's weights are randomly initialized and semantic representations are essentially noise. In our architecture, representation collapse is prevented by the standard positional embeddings added to the patch tokens. At initialization, these positional encodings dominate the cosine similarity calculation, causing the MSN module to behave functionally as a spatial nearest-neighbor search. This provides a safe, structurally sound "warm-start" inductive bias. As training progresses and the network weights diverge from their initialization, the learned semantic features naturally begin to dominate the spatial embeddings, seamlessly shifting the neighbor selection from strictly spatial to deeply semantic.

**Proposition 3 (Robust Neighbor Selection).** *Assume that the dot product similarity between semantically related patches is $\mu_{pos}$ and unrelated patches is $\mu_{neg}$, with $\mu_{pos} > \mu_{neg}$. If the representations are corrupted by isotropic Gaussian noise $\mathcal{N}(0, \sigma^2 \mathbf{I})$, the probability that MSN incorrectly selects a negative patch over a positive patch decays exponentially with the embedding dimension $D$.*

*Proof.* Let $s_{pos} = \mathbf{x}^\top \mathbf{p}$ and $s_{neg} = \mathbf{x}^\top \mathbf{n}$ be the similarity scores for a positive neighbor $\mathbf{p}$ and a negative neighbor $\mathbf{n}$. In the presence of noise, these scores are random variables distributed as $S_{pos} \sim \mathcal{N}(\mu_{\text{pos}}, 2\sigma^2/D)$ and $S_{neg} \sim \mathcal{N}(\mu_{\text{neg}}, 2\sigma^2/D)$ (assuming normalized vectors). The probability of a ranking error (selecting $\mathbf{n}$ instead of $\mathbf{p}$) is:

$$P(\text{Error}) = P(S_{neg} > S_{pos}) = P(S_{neg} - S_{pos} > 0). \tag{15}$$

Let $Z = S_{neg} - S_{pos}$. Then $Z \sim \mathcal{N}(\mu_{\text{neg}} - \mu_{\text{pos}}, \frac{4\sigma^2}{D})$. Using the tail bound for the Gaussian distribution (Chernoff bound):

$$P(Z > 0) \leq \exp\left( -\frac{(\mu_{\text{pos}} - \mu_{\text{neg}})^2}{2 \cdot (4\sigma^2/D)} \right) = \exp\left( -\frac{D(\Delta\mu)^2}{8\sigma^2} \right). \tag{16}$$

This result shows that as the embedding dimension $D$ increases, the probability of selecting incorrect semantic neighbors vanishes exponentially. This guarantees that for high-dimensional ViT representations ($D = 384$ or $D = 768$), the set $\mathcal{P}_i$ used for aggregation almost surely contains only semantically consistent patches, validating the premise of Theorem 1. ∎

## D   Experimental Analyses

In this section, we provide extended experimental results to further validate the design choices and robustness of DMT-JEPA. First, we present a standardized comparison against other masked image modeling and distillation methods such as iBOT and BEiT, demonstrating that our approach remains competitive even without the use of external pre-trained teachers like CLIP. Second, we conduct ablation studies on the Local Aggregation Target module, confirming that the cross-attention mechanism is essential for preserving feature discrepancy compared to simple pooling strategies. Third, we analyze the impact of target discriminability by incorporating view-level and contrastive-level objectives, showing that stronger semantic targets correlate with better linear probing accuracy. Finally, we explore the sensitivity of the Masked Semantic Neighboring module to the number of neighbors and dense pairs, establishing that a local $3 \times 3$ neighborhood with 4 dense pairs strikes the optimal balance between semantic consistency and noise reduction.

**Note on Baseline Omissions:** The standard I-JEPA baseline results for ViT-H on DAVIS and end-to-end fine-tuning on ImageNet are omitted due to prohibitive computational constraints required for large-scale pre-training and exhaustive fine-tuning sweeps. We default to comparing against MAE and DINO at this scale, as their official fine-tuned weights and metrics are publicly available. For example, our method achieved a stable +1.4 mIoU gain ($49.0 \pm 0.12$) on ADE20K semantic segmentation, +1.7 gain ($58.3 \pm 0.14$) on DAVIS video segmentation, and +1.0 AP$^{\text{box}}$ ($50.9 \pm 0.11$) on COCO object detection over I-JEPA.

Table D.1: **Benchmarks with mask modeling methods.** We perform downstream tasks on ImageNet-1K pre-trained ViT-B/16 models on diverse benchmarks to evaluate the quality of fine-tuned representations. The best results are indicated in **bold**.

| Method | mIoU | AP$^{box}$ | AP$^{mask}$ | Clevr/Count | Clevr/Dist | Linear Prob | Finetune |
|---|---|---|---|---|---|---|---|
| BEiT (Bao et al., 2021) | 47.1 | 49.8 | 44.4 | 82.5 | 70.2 | 56.7 | 83.4 |
| MAE (He et al., 2021) | 48.1 | 50.3 | 44.9 | 86.6 | 70.8 | 68.0 | 83.6 |
| iBOT (Zhou et al., 2022) | 50.0 | 51.2 | 44.2 | 81.2 | 70.3 | 79.5 | 84.0 |
| I-JEPA (Assran et al., 2023) | 47.6 | 49.9 | 44.5 | 82.2 | 70.7 | 72.9 | 83.5 |
| DMT-JEPA (ours) | **49.0** | **50.9** | **45.6** | **83.5** | **71.1** | **73.8** | **84.6** |

Table D.2: **Benchmarks with distillation modeling methods.** We perform downstream tasks on ImageNet-1K pre-trained ViT-B/16 models on diverse benchmarks to evaluate the quality of fine-tuned representations. The best results are indicated in **bold**.

| Method | mIoU | AP$^{box}$ | AP$^{mask}$ | Linear Prob | Finetune |
|---|---|---|---|---|---|
| DTM (Kim et al., 2024) | 53.1 | - | - | 77.6 | 85.4 |
| dBOT (Liu et al., 2023) | 49.5 | 52.7 | 45.7 | - | 84.5 |
| Hybrid Distill (Shi et al., 2023) | **49.1** | 50.3 | 44.2 | - | 83.7 |
| DMT-JEPA (ours) | 49.0 | **50.9** | **45.6** | **73.8** | **84.6** |

**Standardization of Baseline Methods.** In Table D.1, we expanded our experimental results using ViT-B/16 to include direct comparisons with both iBOT (Zhou et al., 2022) and BEiT (Bao et al., 2021) across all relevant tasks, thus providing a more consistent evaluation framework and demonstrating the versatility of DMT-JEPA. We also included detailed comparisons with DTM, dBOT, and Hybrid Distill using ViT-B/16 for a more comprehensive evaluation. The quantitative results are reported in Table D.2. Note that both DTM (Kim et al., 2024) and dBOT (Liu et al., 2023) use pre-trained CLIP as a teacher to train the backbone ViT-B/16 on ImageNet-1K, which makes the comparison fully unfair. However, our DMT-JEPA outperforms Hybrid Distill (Shi et al., 2023) without using CLIP regarding most downstream tasks, especially on fine-tuning and object detection.

**Ablation on Cross-attention vs Pooling in Local Aggregation Head.** To validate the effectiveness of using cross-attention layers, we ablated the layer using average-pooling and max-pooling operators. The comparison results are reported in Table D.3. As can be observed, replacing the cross-attention layer with average-pooling deteriorates the results by $1.9@(\mathcal{J}\&\mathcal{F})_m$, $1.4@\mathcal{J}_m$, and $2.4@\mathcal{F}_m$. This might be because average pooling leads to collapsing and losing discrepancy across patch tokens. Our key is each query patch has distinct neighborhood patches, and objectively derived from the distinct neighborhoods will ensure their distinct target objectives. Meanwhile, using the max-pooling operator highly decreases the results in terms of all metrics. These results validate the importance of cross-attention layers in preventing losing discrepancy for distinct target representations during self-supervised pre-training.

**Types of Local Aggregation Heads.** Local aggregation heads affect the ability of the proposed method to aggregate dense targets and context with local semantics. To explore such effects more comprehensively, we varied the type of Local Aggregation Heads from cross-attention and self-attention operators asymmetrically. We report the comparison results on the DAVIS benchmark with ViT-S/16 in Table D.4. When both the context and target head use the cross-attention operators, we achieve the best performance in terms of all metrics. Replacing cross-attention operators with self-attention operators significantly worsens the results in terms of all metrics. These results indicate the difficulty in the asymmetric use of cross-attention aggregation heads, as the target aggregation head is updated using an exponential moving average of the context head weights, and it cannot be solely trained on its architecture.

**Ablation on Discriminative Masked Targets.** To further explore the impact of using different target representations for pre-training on image classification performance, we evaluate models using a linear prob-

Table D.3: **Ablation studies on local aggregation head.** We perform ablation studies on the cross-attention layers in the Local Aggregation Target module using ViT-B/16 pre-trained models. The best results are indicated in **bold**.

| Local Aggregation Head | $(\mathcal{J}\&\mathcal{F})_m$ | $\mathcal{J}_m$ | $\mathcal{F}_m$ |
|---|---|---|---|
| Average-pooling | 55.2 | 54.3 | 56.1 |
| Max-pooling | 55.6 | 54.6 | 56.6 |
| Cross-attention | **57.1** | **55.7** | **58.5** |

Table D.4: **Ablation studies on aggregation head.** We perform ablation studies on context and target aggregation heads in Local Aggregation Target modules using two types (Cross-attention & Self-attention). The best results are indicated in **bold**.

| Context Head $h_\theta$ | Target Head $h_{\tilde{\theta}}$ | $(\mathcal{J}\&\mathcal{F})_m$ | $\mathcal{J}_m$ | $\mathcal{F}_m$ |
|---|---|---|---|---|
| Cross-attention | Cross-attention | **57.1** | **55.7** | **58.5** |
| Self-attention | Cross-attention | 53.7 | 52.5 | 54.9 |
| Self-attention | Self-attention | 50.6 | 49.5 | 51.7 |

ing setup on the ImageNet-1K benchmark and report the top-1 accuracy across different target choices in Table D.5. The results include models trained with two different backbones: ViT-S/16 and ViT-B/16, each trained for 600 epochs. The baseline without customized targets achieves 64.3% accuracy with ViT-S/16 and 72.9% with ViT-B/16. To be specific, we consider two additional approaches, named View-level and Contrastive-level, inspired by EsViT (Li et al., 2022) and PQCL (Zhang et al., 2023), respectively. We note that both approaches can further enhance the discriminative capability of the target representation. When adding a View-level component to Patch-level targets, the performance improves to 65.9% for ViT-S/16 and 73.6% for ViT-B/16. Moreover, adding a Contrastive-level component to the aforementioned targets further enhances the performance, yielding the highest accuracy of 67.2% with ViT-S/16 and 74.5% with ViT-B/16. These results indicate that increasing the discriminative power of the target representation can further enhance the quality of learned visual features, with the combination of all three approaches achieving the best performance across both model variants.

**Impact of Neighbors and Dense Pairs.** To explore the impact of neighbors in neighboring and the number of selected dense pairs, we ablated the neighbors from $\{3 \times 3, 5 \times 5, \text{All patches}\}$ and varied the number of dense pairs from $\{1, 2, 4, 8\}$. The quantitative results on the DAVIS benchmark with ViT-S/16 are reported in Table D.6. As shown in the table, the proposed DMT-JEPA achieved the best performance of $(\mathcal{J}\&\mathcal{F})_m$ when we use $3 \times 3$ neighbors and 4 dense pairs. With the increased number of dense pairs from 1 to 4, the proposed method consistently increases performance as more semantically similar target pairs are extracted. Nevertheless, increasing the number of dense pairs from 4 to 8 will not continually improve the results since 4 dense pairs might be enough to extract the learned dense representations using ViT-S/16. Furthermore, replacing $3 \times 3$ neighbors with all patches significantly deteriorates the performance of all metrics. These results indicate the importance of selecting semantically meaningful neighboring patches for capturing local semantics.

# E    Qualitative Analyses

In this section, we present additional analyses on the effectiveness of cross-attention layers in the local aggregation head, and computational comparison with I-JEPA (Assran et al., 2023). In order to qualitatively demonstrate the effectiveness of the proposed DMT-JEPA in learning local semantics during pre-training, we provide learned attention maps from the target encoder using pre-trained I-JEPA and our method, and attention maps from the cross-attention layer in the proposed Local Aggregation Target (LAT) modules in Figure E.1, E.2, and E.3. Furthermore, we visualize the top-10% of highly correlated patches in more examples by thresholding the cosine-similarity maps of query patches in the last layer in Figure D.1, and D.2.

Table D.5: **Ablation studies on discriminative masked** targets. We perform a linear evaluation on pre-trained ViT-S/16 and ViT-B/16 models for image classification on ImageNet-1K benchmark. We report the top-1 accuracy to evaluate the quality of pre-trained representations with discriminative masked targets. The best results are indicated in **bold**.

| Patch-level | View-level | Contrastive-level | Backbone | Epochs | Top-1 Acc |
|:---:|:---:|:---:|:---:|:---:|:---:|
| ✗ | ✗ | ✗ | | 600 | 64.3 |
| ✓ | ✓ | ✗ | S/16 | 600 | 65.9 |
| ✓ | ✓ | ✓ | | 600 | **67.2** |
| ✗ | ✗ | ✗ | | 600 | 72.9 |
| ✓ | ✓ | ✗ | B/16 | 600 | 73.6 |
| ✓ | ✓ | ✓ | | 600 | **74.5** |

Table D.6: **Ablation studies on hyperparameters.** We perform ablation studies using a pre-trained ViT-S/16 model to explore effects on neighbors and dense pairs in the Masked Semantic Neighboring module. The best results are indicated in **bold**.

| Neighbors | # Dense Pairs | $(\mathcal{J}\&\mathcal{F})_m$ | $\mathcal{J}_m$ | $\mathcal{F}_m$ |
|:---:|:---:|:---:|:---:|:---:|
| 3×3 | | **57.1** | **55.7** | **58.5** |
| 5×5 | 4 | 56.3 | 54.9 | 57.7 |
| All patches | | 48.5 | 46.7 | 50.3 |
| | 1 | 55.9 | 54.3 | 57.5 |
| 3×3 | 2 | 56.3 | 54.9 | 57.7 |
| | 8 | 56.7 | 55.3 | 58.1 |

**Learned Attention Maps.** Learning discriminative attention maps is one of the key aspects of capturing local semantics for downstream tasks of dense prediction type, such as segmentation and detection. To better evaluate the quality of learned attention maps, we visualize the averaged maps from different heads in the last attention layer by using a pre-trained ViT-B/16 target encoder. For a more comprehensive comparison, we also add the attention maps from I-JEPA (Assran et al., 2023) target encoder and the cross-attention layer in our Local Aggregation Target (LAT) module. The qualitative visualization results are shown in Figure E.1, E.2, and E.3. Note that columns for each image sample represent the original image, attention maps from the target encoder in I-JEPA, attention maps from the target encoder in our DMT-JEPA, and attention maps from the local aggregation target module in our DMT-JEPA. As can be seen, both attention maps from the target encoder and the local aggregation target module in our DMT-JEPA are discriminative and focus on the object semantics in the image. However, the attention maps from the target encoder in I-JEPA activate both the foreground and background and can not effectively discriminate the object semantics, because they did not incorporate local semantics into target representations as our DMT-JEPA did. Meanwhile, the attention maps of the local aggregation target module in our DMT-JEPA have more focus on the details of foreground objects than that from the target encoder, indicating the effectiveness of our local aggregation target module in generating target representations with local semantics. These high-quality visualization results further demonstrate the superiority of our new framework in learning meaningful representations with local semantics during self-supervised training, compared to I-JEPA, the state-of-the-art image-based joint-embedding predictive architecture.

**Learned Cosine Similarity Maps.** To further validate the effectiveness of our method in learning discriminative dense representations, we visualize the top-10% of highly correlated patches by thresholding the cosine similarity maps of query patches in the last layer using the pre-trained ViT-B/16 target encoder. Figure D.1 and D.2 showcase the qualitative visualization results, where rows for each sample denote the location of the given query patch and the top-10% patches. We can observe that the patches extracted by our DMT-JEPA are centralized and specifically focus on the location of the given query patch. For example, given the first query patch on the building in the car example shown in Figure D.1, the top-10% patches focus on the building. When the query patch is given on the location of the car, the top-10% patches also

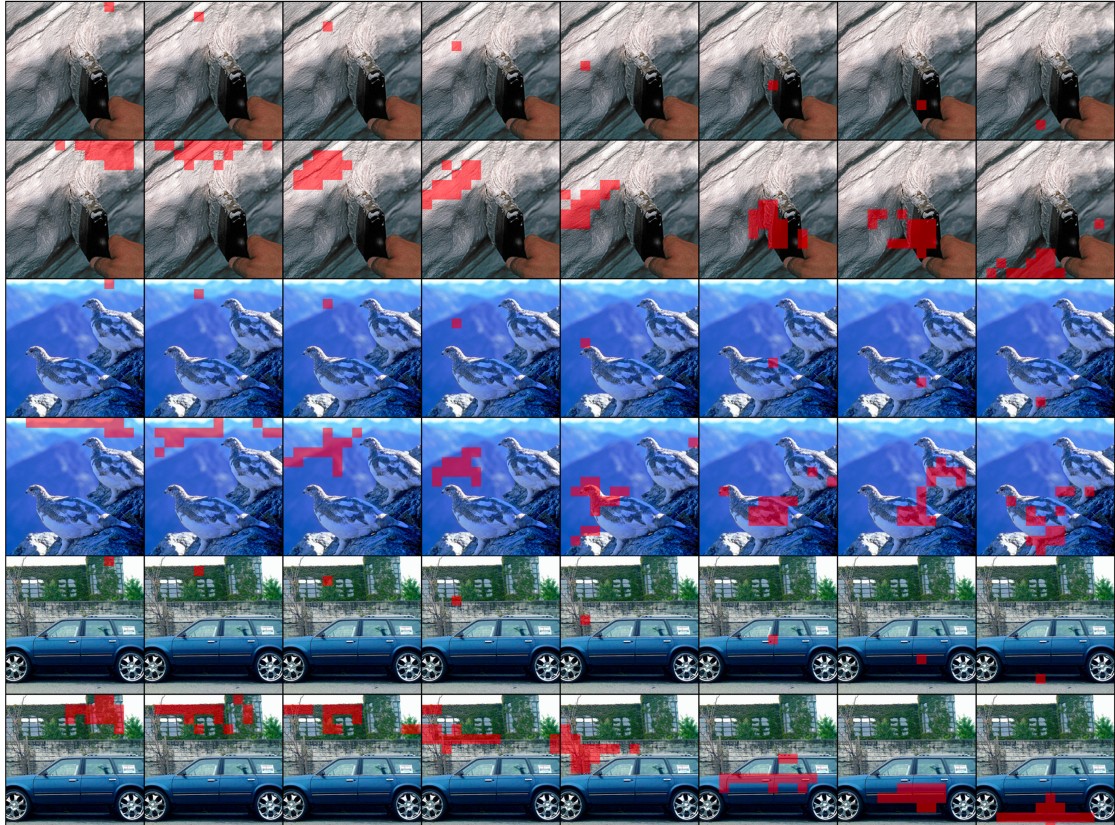

Figure D.1: Qualitative visualization of learned cosine similarity maps given query patches using ViT-B/16. Rows for each sample denote the location of the given query patch and the top-10% patches. Our DMT-JEPA performs effectively by encouraging the model to learn local semantics.

change to focus on the car. Another example can also be seen in the dynamic changes with respect to the location of query patches on the head, arm, elbow, and microphone in the first sample shown in Figure D.2. Interestingly, when the query patch is given on the body part from one of two birds in the second example shown in Figure D.1, the top-10% patches can highlight the location of body parts from both birds, which might be due to the similar local semantics in both body locations. These visualization results demonstrate that our DMT-JEPA performs effectively by encouraging the model to learn discriminative representations.

# F    More Discussions

## F.1    Broader Impact

Our method contributes to the field of self-supervised learning, enabling high-performance visual representations without the need for expensive human annotations. This has positive implications for democratizing AI in data-scarce domains (*e.g.*, medical imaging, satellite monitoring). However, as with all foundation models trained on large-scale web data (ImageNet), there is a risk of inheriting biases present in the training set. Future work should investigate the fairness of representations learned via local semantic aggregation. Furthermore, while efficient, the training of large ViT models entails a carbon footprint; our method's efficiency improvements over pixel-reconstruction methods help mitigate this environmental cost.

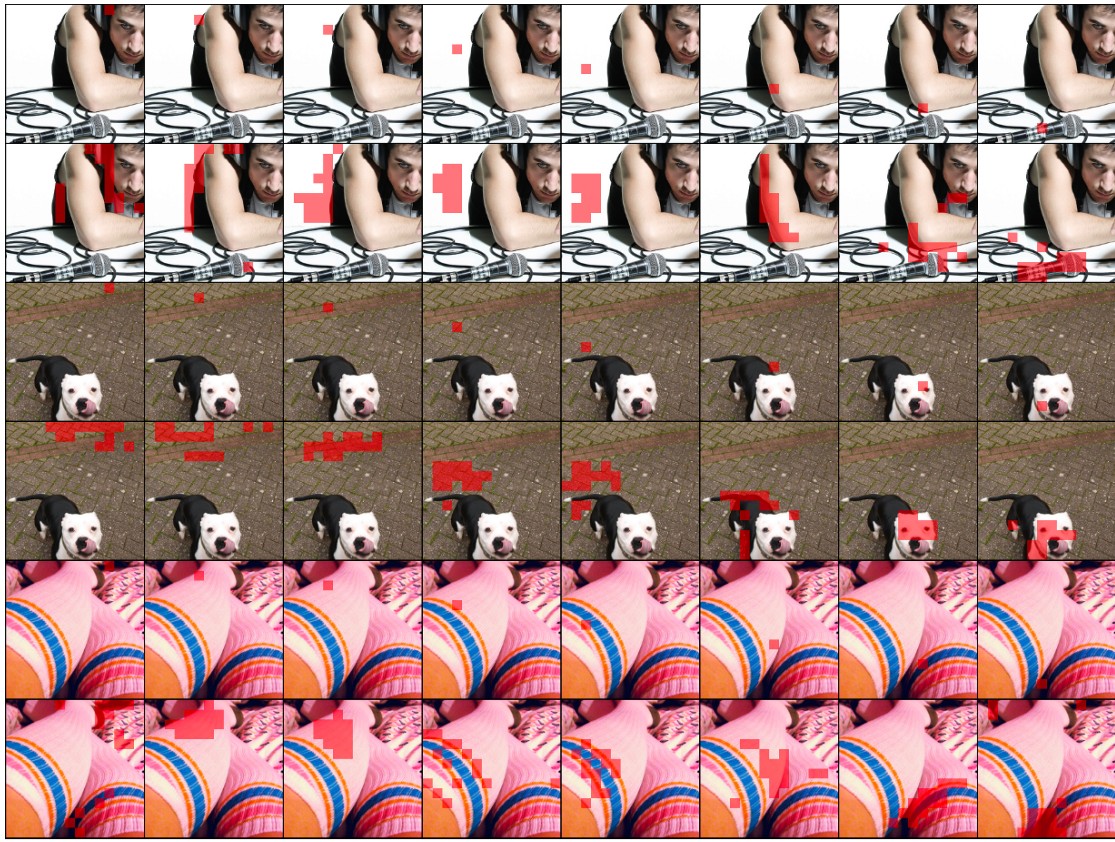

Figure D.2: Qualitative visualization of learned cosine similarity maps given query patches using ViT-B/16. Rows for each sample denote the location of the given query patch and the top-10% patches. Our DMT-JEPA performs effectively by encouraging the model to learn local semantics.

## F.2 Limitations

While DMT-JEPA improves over I-JEPA, it introduces a small computational overhead due to the neighbor search and aggregation step (Masked Semantic Neighboring). Although our experiments show this cost is negligible compared to the total training time, it may scale linearly with neighborhood size if not optimized. Additionally, in highly textured or homogeneous regions (*e.g.*, sky, ocean), finding "semantically distinct" neighbors may be trivial or noisy, potentially providing less informative gradients compared to edge-rich regions.

**Limitations in Homogeneous Regions:** A primary limitation of DMT-JEPA is its reliance on informative local neighborhoods. In highly homogeneous or perfectly repetitive visual regions (*e.g.,* clear skies, blank walls), the neighboring patches provide no distinct semantic variance. In such cases, the Local Aggregation Target (LAT) simply averages near-identical vectors, offering no meaningful variance reduction or semantic enrichment over a single-patch target. From an information-theoretic perspective, these regions possess near-zero local entropy. Because the mutual information between adjacent patches is maximized, the cross-attention mechanism within the LAT module encounters uniformly high key-query similarities, often leading to a degenerate, uniform attention distribution. Consequently, the gradient signal derived from predicting this aggregated target provides minimal useful supervisory feedback for learning complex semantic abstractions. While the architecture's momentum encoder safely prevents localized representation collapse, continuously optimizing over these zero-variance neighborhoods acts as a computational inefficiency that could decelerate overall convergence.

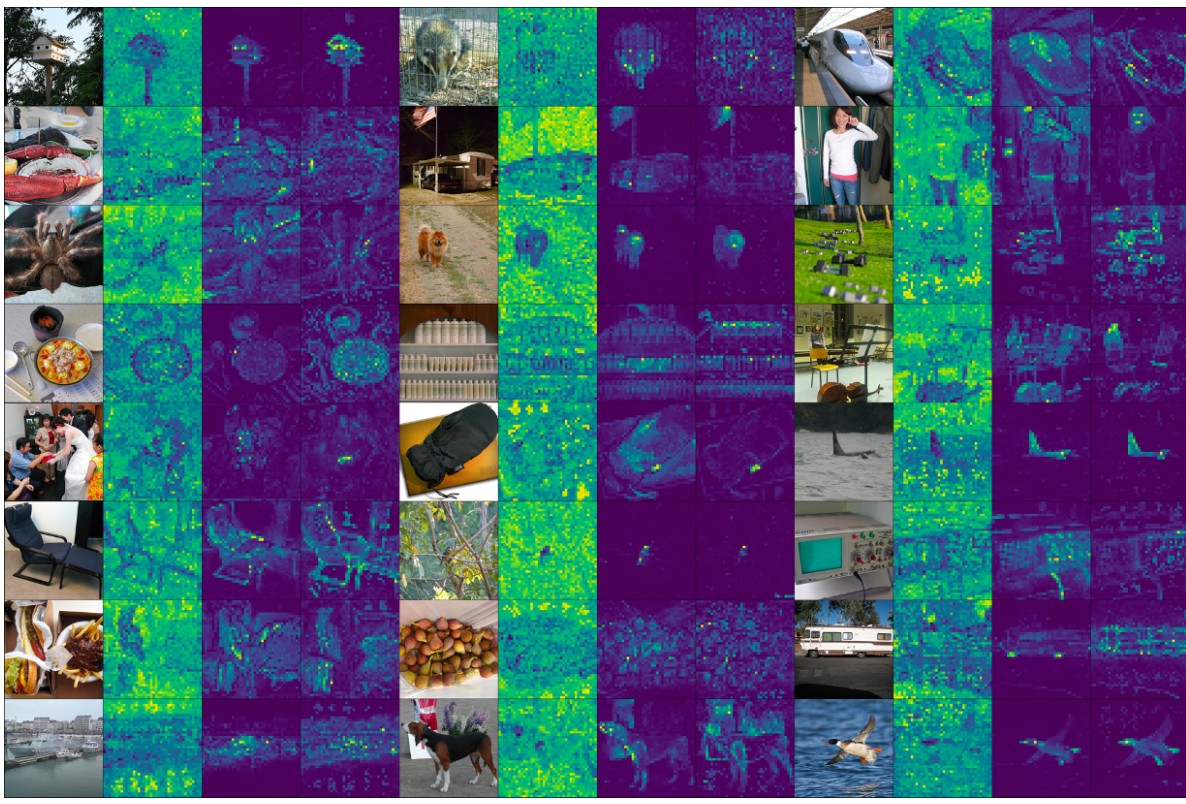

Figure E.1: Qualitative visualization of learned attention maps using ViT-B/16 model. Columns for each sample denote the original image, attention maps from target encoder in I-JEPA Assran et al. (2023), attention maps from target encoder in our DMT-JEPA, and attention maps from the local aggregation head in our DMT-JEPA. Our DMT-JEPA achieves much better attention maps.

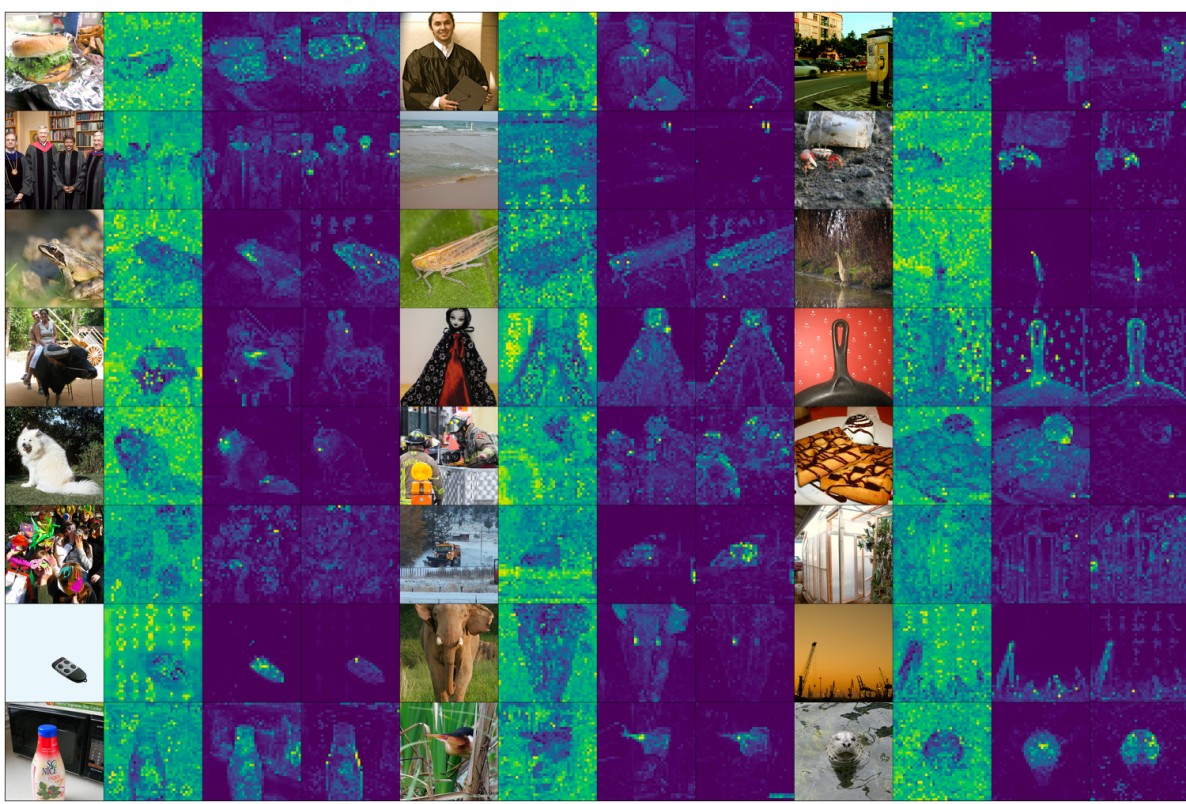

Figure E.2: Qualitative visualization of learned attention maps using ViT-B/16 model. Columns for each sample denote the original image, attention maps from target encoder in I-JEPA Assran et al. (2023), attention maps from target encoder in our DMT-JEPA, and attention maps from the local aggregation head in our DMT-JEPA. Our DMT-JEPA achieves much better attention maps.

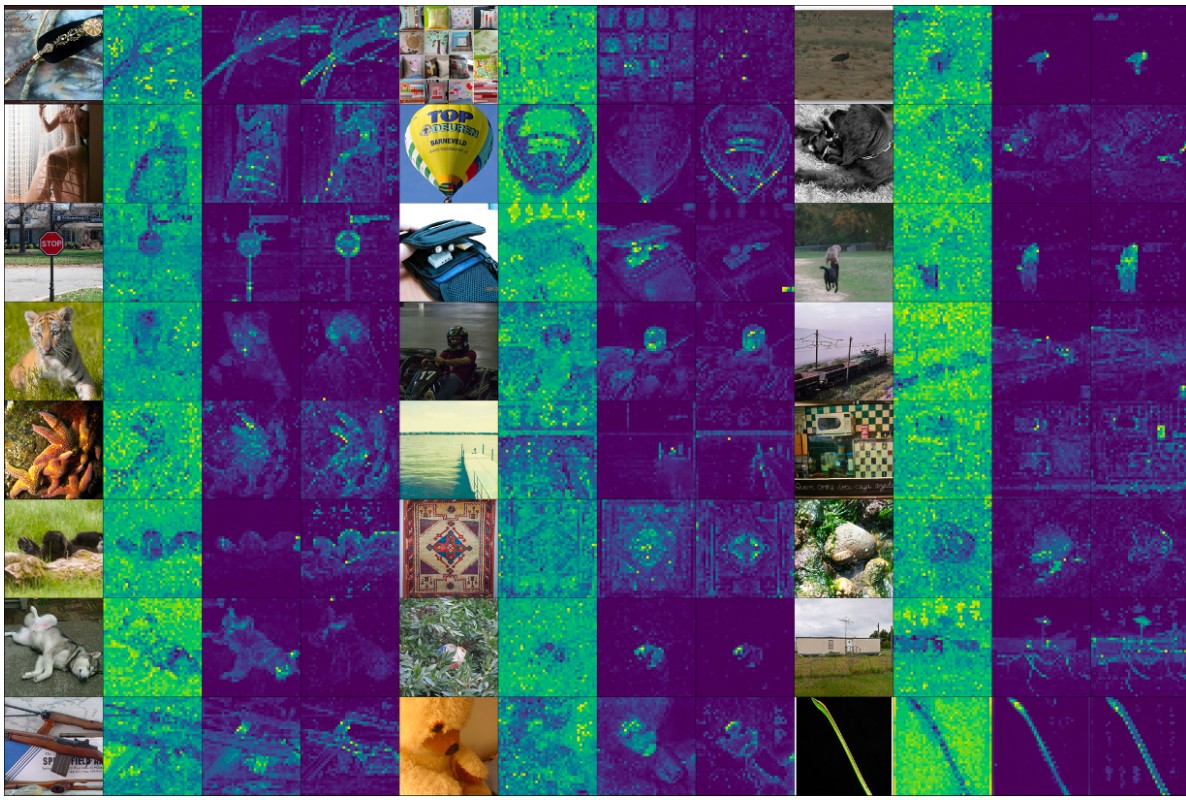

Figure E.3: Qualitative visualization of learned attention maps using ViT-B/16 model. Columns for each sample denote the original image, attention maps from target encoder in I-JEPA (Assran et al., 2023), attention maps from target encoder in our DMT-JEPA, and attention maps from the local aggregation head in our DMT-JEPA. Our DMT-JEPA achieves much better attention maps.

