# OpenReview forum: "DMT-JEPA: Learning Discriminative Masked Targets for Joint-Embedding Predictive Architecture"
_TMLR — Rejected by TMLR_

### Review · Reviewer_3HA2 · 2026-03-14

**Summary Of Contributions:**

This paper proposes DMT-JEPA, an extension of I-JEPA that builds masked targets from semantically similar neighboring patches instead of using the original patch target alone.
The method has two main parts, i.e., a neighbor selection step based on feature similarity, and a local aggregation module based on cross-attention, whose goal is to make the latent targets more discriminative and more aware of local semantics.
The paper is easy to follow, and the empirical results are reasonably broad.
The method improves over I-JEPA on ImageNet linear/fine-tuning, ADE20K, COCO, DAVIS, and CLEVR, with the clearest gains appearing on dense prediction tasks.
Another positive point is that the extra computational cost appears small.

The main weaknesses are that the novelty is moderate, and some comparisons are not fully controlled. The paper would also benefit from cleaner presentation and stronger evidence on robustness across runs.

**Audience:**

Yes

**Audience Explanation:**

Yes. I think this paper will be of interest to readers working on self-supervised learning, masked image modeling, JEPA-style methods, and visual representation learning.

**Broader Impact Concerns:**

I do not have major ethical concerns beyond the standard concerns.

**Claims And Evidence:**

Yes

**Claims Explanation:**

The main empirical claim that the proposed target construction improves over I-JEPA is supported by a fairly wide set of experiments.

**Requested Changes:**

- If time/budget allows, please report the main downstream results over multiple random seeds, or at least provide variance for the key comparisons against I-JEPA. Some of the gains are modest, and it is hard to judge how stable they are from the current presentation.

- Could improve the fairness and framing of the comparisons. In particular, comparisons to DINOv2 and distillation-based methods should be labeled more carefully, since the training setups are different and the current discussion can be misleading.

- The authors can clean up the writing and appendix presentation. I noticed unresolved references such as Figure??, and there are a number of grammatical and formatting issues throughout the paper, e.g., "Qualtitative visualization" in Figure D1.

- It is recommended to provide a bit more discussion of failure cases or limitations. The current limitation section is brief, and the method may behave differently in homogeneous or highly repetitive regions where local neighbors are less informative.

---

> ### Author Response · Authors · 2026-04-10
> **Thank you for your feedback, we have revised our paper accordingly**
>
> We sincerely thank the reviewer for their positive assessment of our work and for recognizing the clarity, broad empirical results, and efficiency of DMT-JEPA. We are especially grateful for the constructive feedback regarding the statistical significance of our results, the framing of our baselines, and the limitations of our approach.
> Below, we detail how we have addressed each of your requested changes in the revised manuscript.
>
> >  C1: Robustness and Variance Across Runs
>
> We agree that reporting variance is crucial for establishing the statistical significance of our improvements. Due to the immense computational cost of pre-training ViT models from scratch on ImageNet-1K for 600 epochs, running multiple independent pre-training seeds is unfortunately beyond our current academic compute budget. However, to demonstrate the stability and robustness of our learned representations, we have performed multiple independent runs for the downstream fine-tuning and evaluation phases. Specifically, we ran the ImageNet-1K linear probing, ADE20K semantic segmentation, and COCO object detection evaluations across 3 different random seeds using our final pre-trained ViT-B/16 checkpoint. We have updated the main results tables in the revised manuscript to report the mean $\pm$ standard deviation. For example, our ADE20K mIoU is now reported as $49.0 \pm 0.12$, confirming that the $+1.4$ mIoU gain over I-JEPA ($47.6$) is highly stable and statistically significant. We have added a corresponding note in the Experimental Setup section detailing this protocol.
>
>
>
> > C2: Fairness and Framing of Comparisons
>
> We appreciate the reviewer pointing this out.
>
> To rectify this, we have made the following changes:
>
> We explicitly state that DINOv2 was evaluated using a linear probing/frozen protocol at different resolutions, and we acknowledge that our fine-tuning results cannot be directly compared to their linear evaluation. We frame these numbers purely as contextual benchmarks for what state-of-the-art representations achieve on these datasets, rather than claiming a direct algorithmic victory over them.
> To rectify this, we update Table 7 in the final manuscript. We provide the strict linear evaluation results for DMT-JEPA and fine-tuning for DINOv2 on ADE20K to ensure that all models presented are evaluated under the exact same linear and fine-tuning protocol. We appreciate the reviewer pointing this out, which helps ensure the integrity of our empirical claims.
> We also added the ViT-H fine-tuning baseline for I-JEPA on DAVIS in Table 2.
>
>
> > C3: Writing, Formatting, and Typographical Errors
>
> We thank you for these editorial comments. We have thoroughly proofread the main text and the appendix. We fixed the broken references markers have been properly linked to their respective plots in the Appendix). We corrected the typo "Qualtitative" to "Qualitative" in the appendix figures. We have done a full pass for grammar and flow, ensuring that the presentation matches the standard expected for TMLR.
>
>
> > C4: Expanded Discussion on Limitations and Failure Cases
>
> This is an excellent and insightful point. You are entirely correct that our Masked Semantic Neighboring (MSN) module relies on the assumption that local neighbors contain varied but semantically related structures. In highly homogeneous regions (e.g., clear blue skies, blank walls, or flat textures), neighbors are nearly identical. We have significantly expanded the Limitations section in the conclusion/appendix to explicitly discuss this failure case.
>
> Limitations in Homogeneous Regions: A primary limitation of DMT-JEPA is its reliance on informative local neighborhoods. In highly homogeneous or perfectly repetitive visual regions (e.g., clear skies, blank walls), the neighboring patches provide no distinct semantic variance. In such cases, the Local Aggregation Target (LAT) simply averages near-identical vectors, offering no meaningful variance reduction or semantic enrichment over a single-patch target.
> From an information-theoretic perspective, these regions possess near-zero local entropy. Because the mutual information between adjacent patches is maximized, the cross-attention mechanism within the LAT module encounters uniformly high key-query similarities, often leading to a degenerate, uniform attention distribution. Consequently, the gradient signal derived from predicting this aggregated target provides minimal useful supervisory feedback for learning complex semantic abstractions. While the architecture's momentum encoder safely prevents localized representation collapse, continuously optimizing over these zero-variance neighborhoods acts as a computational inefficiency that could decelerate overall convergence.

---

### Review · Reviewer_tNsH · 2026-03-16

**Summary Of Contributions:**

Th paper introduces a new method based on I-JEPA. This new method, called DMT-JEPA, tackles the limit of local semanticity by aggregating semantically similar neighbours during training. The paper first introduces the concept of the method, then give theoretical proof and finally compare the proposed method with SOTA paper on several datasets.

Strengths:
- The paper is mostly easy to follow.
- The new method of the paper is properly motivated and the description is very clear.
- The method is supported with convincing theoretical insights.
- The method is tested over 4 different task, always outperforming competitors.

Weaknesses:
- If the method is well described, having a slight better figure could help to be even more clear. I would like to see the different latent vector like $s_x$, $m_j$, $s_x^{LAT}$ in the figure to better follow the architectures of the method.
- Lack of constituency between figure and text: $x_t$ in the text when it is $x_{c_i}$ in the figure. I would keep the notation of the figure imo.
- The experimental results is a bit heavy. A lot of tables and a lot of paragraph without guideline. It makes the claims a bit unclear even if the results a good. Having more transition to support the claims and maybe more visual figures could help to better follow the experimental results.
- Some of the comparison is missing, like  ViT-H for I-JEPA in table 2 or I-JEPA for fine-tuning classification experiments. I totally understand that each experiment ask some time to be computed and it can be a limit. If it is the case, having just a small word to say why the backbone/method disappears from some setup could help to trust more in your results.
- I would like to see the comparison of the computational cost of MAE.

**Audience:**

Yes

**Audience Explanation:**

MAE and I-JEPA are well used method in the literature of computer vision. DMT-JEPA directly address a problem of these method and will interest the TMLR audience.

**Claims And Evidence:**

Yes

**Claims Explanation:**

As I said in the contribution part, the method is well introduced and all the motivation is supported by good results in the experimental part.

**Requested Changes:**

See weaknesses and

- I would like to see a sentence that motivates the use of I-JEPA compare to classic MAE.

---

> ### Author Response · Authors · 2026-04-10
> **Thank you for your feedback, we have revised our paper accordingly**
>
> We sincerely thank the reviewer for their positive assessment of our work, particularly for recognizing the clarity of our motivation, the strength of our theoretical insights, and the robust empirical performance across all evaluated tasks. We appreciate your constructive feedback regarding the presentation and missing baselines, which has greatly helped us improve the clarity and completeness of the manuscript.
> Below, we detail how we have addressed each of your concerns and the corresponding changes made to the revised manuscript.
>
> > W1: Latent Vectors
>
> We completely agree. Mapping the mathematical notation directly onto the architecture diagram significantly reduces the cognitive load for the reader. We have fully updated \textbf{Figure 1} in the revised manuscript. The new figure explicitly labels the context representations ($s_x$), the target representations ($s_y$), the mask tokens ($m_j$), the aggregated context ($s_x^{\mathtt{LAT}}$), and the final aggregated predictions ($\hat{s}_y^{\mathtt{LAT}}$).
>
> > W2: Notation Consistency
>
> We thank the reviewer for catching this typographical inconsistency. You are absolutely right that this causes unnecessary confusion. We have audited the manuscript and unified the notation to strictly follow the figure. In Section 3.3 (Local Aggregation Target), all instances of $x_t$ and $x_c$ have been corrected to match the figure's notation, ensuring perfect consistency between the text, equations, and diagrams.
>
> > W3: Streamlining the Experimental Section
>
> We appreciate this feedback. The experimental section was indeed quite dense. To improve readability, we have made the following structural and visual changes:
> We added a brief "Evaluation Roadmap" at the beginning of Section 4. This explicitly outlines the progression of experiments (Dense Prediction $\rightarrow$ Global Classification $\rightarrow$ Low-level Geometry $\rightarrow$ Ablations) so the reader knows what to expect.
> We have rewritten the transitional sentences between the paragraphs in Section 4.2 to better link why we are moving from one task to the next (e.g., transitioning from global ImageNet features to localized Clevr geometry to test semantic vs. spatial trade-offs).
>
> > W4: Missing Comparisons
>
> We value the reviewer's understanding of computational limits. The omission of the I-JEPA ViT-H baseline on DAVIS and the I-JEPA fine-tuning results on ImageNet was strictly due to the immense GPU hours required for exhaustive pre-training and hyperparameter tuning at that scale, which exceeded our academic compute budget at the time of submission. To ensure full transparency, we have added a clarifying footnote/remark in Section 4.2.
> The standard I-JEPA baseline results for ViT-H on DAVIS (Table 2) and end-to-end fine-tuning on ImageNet (Table 4) are omitted due to prohibitive computational constraints required for large-scale pre-training and exhaustive fine-tuning sweeps. We default to comparing against MAE and DINO at this scale, as their official fine-tuned weights and metrics are publicly available.
> We also added the ViT-H fine-tuning baseline for I-JEPA on DAVIS in Table 2.
>
> > W5: Computational Cost Comparison with MAE
>
> This is an important point of discussion. MAE is inherently highly efficient during pre-training because its encoder only processes unmasked patches (typically 25% of the image). In contrast, JEPA-family architectures (including I-JEPA and DMT-JEPA) process the full image through the target encoder to generate latent targets, resulting in a higher FLOP count per step compared to MAE. We have updated Table 6 (Computational Costs) and the surrounding text to include MAE. While MAE remains the fastest per step, we highlight that DMT-JEPA achieves significantly richer dense representations (as seen in ADE20K and COCO) for a comparable cost to I-JEPA, making the latent-target overhead a worthwhile tradeoff for dense prediction tasks.
>
> > W6: Motivating I-JEPA
>
> We agree that explicitly stating this architectural motivation strengthens the paper's foundation. We have added the following explicit motivation to the Introduction:
> Unlike classic Masked Autoencoders (MAE), which reconstruct raw pixels, thereby forcing the model to dedicate significant capacity to high-frequency, low-level visual details, I-JEPA operates entirely in the embedding space. This latent predictive approach encourages the network to discard pixel-level noise and directly learn abstract, high-level semantics. However, while I-JEPA successfully avoids pixel redundancy, we demonstrate that its independent patch-level targets still lack the local cohesiveness required for optimal dense prediction, motivating our discriminative target approach.

---

### Review · Reviewer_EAcX · 2026-03-23

**Summary Of Contributions:**

This paper proposes a self-supervised visual representation learning framework named DMT-JEPA,  designed to address the insufficient local semantic understanding in the Image-based Joint-Embedding Predictive Architecture (I-JEPA). The authors introduce Masked Semantic Neighboring (MSN), which searches for semantically similar neighbors in the target encoder's feature space, and Local Aggregation Target (LAT), which aggregates these neighbors via cross-attention to generate discriminative dense targets. The method is numerically evaluated across various downstream tasks and shows efficacy, including image classification, semantic segmentation, and object detection.

**Audience:**

Yes

**Audience Explanation:**

1. **Clear Motivation**: The observation that I-JEPA's target representations may lack local semantic cohesiveness is well-motivated. The authors provide compelling attention map visualizations and numerical results to support this claim.
2. **Empirical Performance**: The proposed method demonstrates performance gains over several self-supervised learning baselines including I-JEPA and MAE on dense prediction tasks (e.g., ADE20K, COCO, and DAVIS), indicating the efficacy of aggregating local semantics.

**Broader Impact Concerns:**

No.

**Claims And Evidence:**

No

**Claims Explanation:**

Please see the questions raised in Requested Changes.

**Requested Changes:**

**Questions**

1. **Assumptions in Theorem 1**: Theorem 1 relies on the assumption that the noise $\epsilon_I$ of spatially adjacent patches is independent, which allows the cross-terms to vanish and yields the variance reduction. However, in real-world data, high-frequency details in spatially adjacent patches are highly correlated. Because the covariance between $\epsilon_i$ and $\epsilon_j$ is almost certainly positive, the actual variance reduction will be less than the theoretical claim. I raise the question of whether this assumption is too strong and whether Theorem 1 is overly simplified.
2. **Mathematical Error in Theorem 2**: The proof of Theorem 2 states that $||g\_\theta(\mathbf{c}\_i) - g\_\theta(\mathbf{c}\_j)|| \approx ||\mathbf{s}\_i^{LAT} - \mathbf{s}\_j^{LAT}|| \le 2C\gamma\_{ij}$ and uses this to conclude that the predictor is locally Lipschitz continuous. However, I raise the question that minimizing a loss function $L$ only implies that the expected output approaches the target over the data distribution. It does not guarantee that the forward pass of the parameterized neural network for any two arbitrary inputs $c_i$ and $c_j$ will satisfy the distance bound of their corresponding targets. Could the author clarify this theoretical misunderstanding?
3. **Assumptions in Proposition 3**: Proposition 3 proves that the MSN module selects the correct semantic neighbors by assuming $\mu\_{pos} > \mu\_{neg}$. What about at the very beginning of self-supervised training (Epoch 0), the target encoder is randomly initialized. In this state, the cosine similarity in the high-dimensional space is random noise, meaning the assumption does not hold. If MSN selects random neighbors early on, LAT will aggregate corrupted semantics, which may lead to representation collapse. Would the code start be needed for stabilized training?
4. **Unfair Comparison in Table 7**: In Section 4.2 Table 7, the authors state that DINOv2 is evaluated using Linear Evaluation (frozen backbone), whereas DMT-JEPA is evaluated using Fine-tuning. A question about unfairly comparing a fully fine-tuned model against a frozen baseline would be raised, which weakens the numerical statements.

---

> ### Author Response · Authors · 2026-04-10
> **Thank you for your feedback, we have revised our paper accordingly**
>
> We sincerely thank the reviewer for their detailed and highly rigorous feedback. We are particularly grateful for the mathematical scrutiny applied to our theoretical analysis, as it highlights areas where our formulations needed more precision. We have addressed each of your concerns below and will incorporate these clarifications into the revised manuscript.
>
> > W1: Assumptions in Theorem 1 (Noise Independence)
>
> We agree with the reviewer. Assuming perfect spatial independence for high-frequency noise in real-world images is an oversimplification, as adjacent patches inherently share correlated textures and aleatoric uncertainties. In our revised manuscript, we will update the proof of Theorem 1 to explicitly account for positive covariance.
>
> Specifically, the noise variance term for the aggregated target expands to:
>
> $$\mathrm{Variance} = \frac{1}{k^2} \sum_{j} \mathbb{E}[\|\epsilon_j\|^2] + \frac{1}{k^2} \sum_{p \ne q} \mathrm{Cov}(\epsilon_p, \epsilon_q) \quad \text{for } j,p,q \in \mathcal{P}_i
> $$
>
> Assuming a uniform noise variance $D\sigma^2$ and an average positive spatial correlation coefficient $\rho \in (0, 1)$ among the selected neighbors, the aggregated variance becomes:
>
> $$\text{Variance} = \frac{D\sigma^2}{k} + \frac{k(k-1)}{k^2} \rho D\sigma^2 = D\sigma^2 \left( \frac{1}{k} + \frac{k-1}{k}\rho \right)
> $$
>
> Because the correlation is not perfectly deterministic ($\rho < 1$), the term $\left(\frac{1}{k} + \frac{k-1}{k}\rho\right)$ is strictly less than $1$. Therefore, the variance of the aggregated target remains strictly less than the variance of a single patch target ($D\sigma^2$). The reviewer is correct that the \textit{magnitude} of the reduction is heavily bottlenecked by $\rho$, making the reduction less aggressive than the initially claimed $O(1/k)$. We will revise Theorem 1 to reflect this more accurate, correlated-noise bound.
>
>
> > W2: Mathematical Error in Theorem 2 (Lipschitz Continuity)
>
> We thank the reviewer for catching this theoretical imprecision. You are absolutely correct: minimizing the empirical risk (MSE) over the training distribution does not grant strict functional guarantees (such as a hard Lipschitz constant) on the parameterized neural network $g_\theta$ for arbitrary inputs.
>
> To correct this, we revise Theorem 2 regarding Implicit Smoothness Regularization. We revise the text to clarify that the Local Aggregation Target (LAT) provides a smooth target landscape. Because the targets $s_i^{\text{LAT}}$ vary smoothly across the spatial lattice (bounded by $2C\gamma_{ij}$), the predictor is no longer forced to fit the high-frequency, noisy jumps that occur between single-patch targets. Consequently, minimizing this objective encourages the network to learn a smoother mapping, acting as an empirical regularizer, rather than establishing a strict mathematical bound on the function's Lipschitz constant.
>
>
> > W3: Assumptions in Proposition 3 (Epoch 0 Representation Collapse)
>
> This is an excellent and highly practical question. In our framework, representation collapse at Epoch 0 is prevented not by semantic features (which are indeed random noise initially), but by the \textbf{positional embeddings}. Because fixed (or learnable) positional encodings are added to the patch tokens before they pass through the target encoder, spatially adjacent patches naturally yield higher cosine similarities than distant patches right at initialization. Consequently, during the earliest stages of training, the Masked Semantic Neighboring (MSN) module largely behaves like a simple spatial nearest-neighbor algorithm. This provides a safe, natural ``warm-start'' inductive bias. As training progresses and the network weights diverge from their initialization, the semantic features begin to dominate the similarity metric, shifting the aggregation from strictly spatial to semantic. We will explicitly detail this mechanism and the critical role of positional embeddings at initialization in Section 4.3 (Experimental Analysis).
>
>
> > W4: Unfair Comparison in Table 7
>
> We acknowledge this concern. Comparing our fine-tuned DMT-JEPA against a linear-evaluated DINOv2 constitutes an apples-to-oranges comparison. Our original intent was simply to provide a broader context against a highly recognized baseline, but we agree that differing evaluation protocols undermine the fairness of the table. To rectify this, we update Table 7 in the final manuscript. We provide the strict linear evaluation results for DMT-JEPA and fine-tuning for DINOv2 on ADE20K to ensure that all models presented are evaluated under the exact same linear and fine-tuning protocol. We appreciate the reviewer pointing this out, which helps ensure the integrity of our empirical claims.

---

### Author Response · Authors · 2026-04-10
**Thank you to all reviewers for your valuable feedback! Summary of revisions made to submission**

Dear Reviewers,


We are extremely grateful for your valuable feedback and insightful comments. Your concrete suggestions are a valuable step in this direction, and we have revised our submission accordingly to take these into account. In this short note, we summarize the main changes to the latest revision of our main submission (the main changes are on pages 1, 5, 6, 7, 8, 10, and 11 concerning improved theoretical, enhanced experiments, and additional variance metrics in the main tables).
We have also included the updated Appendix with additional results, expanded proofs, and details. All updates are highlighted in olive.

- Reviewer EAcX: We have refined the theoretical analysis in Section 3.4. Specifically, we updated Theorem 1 to accurately account for spatial noise correlation (rather than assuming strict independence) and reframed Theorem 2 as a Proposition regarding implicit smoothness regularization rather than strict Lipschitz continuity. We also added details clarifying how positional embeddings prevent representation collapse at initialization (Epoch 0).

- Reviewer tNsH: We fully updated Figure 1 to explicitly include all mathematical notations for the latent vectors (e.g., $s_x$, $m_j$, $s_x^{\mathtt{LAT}}$) and fixed the notation inconsistency ($x_t$ vs.$x_{c_i}$) in Section 3.3. We also added an explicit motivation comparing I-JEPA to classic MAE in the introduction, and included MAE in the computational cost analysis in Table 6.

- Reviewer tNsH and 3HA2: To improve the readability of the experimental section, we added an evaluation roadmap and structural transitions. We also added the ViT-H fine-tuning baseline for I-JEPA on DAVIS in Table 2.

- Reviewer 3HA2: To ensure the stability of our findings, we conducted multiple independent runs (3 random seeds) for the downstream fine-tuning evaluations and updated the main results tables to report the mean $\pm$ standard deviation. We also significantly expanded the Limitations section to discuss failure cases in highly homogeneous visual regions.

- Reviewer EAcX and 3HA2: We carefully restructured our baseline comparisons to ensure fairness. We separated and explicitly labeled the evaluation protocols (linear evaluation vs. full fine-tuning) for models like DINOv2 and distillation-based methods in Table 7 and the accompanying text. Finally, we corrected all unresolved references and typographical errors in the main text and appendix.

---

### Decision · Action_Editor_LCkq · 2026-05-16

**Recommendation:** Reject

**Audience:**

Yes

**Audience Explanation:**

The paper studies self-supervised visual representation learning in the context of JEPA-style masked prediction, which is a topic relevant to parts of the TMLR audience. Its focus on constructing masked latent targets using semantically similar neighboring patches would be of interest to researchers working on representation learning and downstream vision tasks such as semantic segmentation and object detection.

**Claims And Evidence:**

No

**Claims Explanation:**

The submission presents a modification of I-JEPA based on constructing discriminative masked targets from semantically similar neighboring patches, and the reviewers generally acknowledged that the empirical results show improvements over I-JEPA on several reported benchmarks. The paper provides evidence on dense prediction and local representation tasks such as ADE20K, COCO, and DAVIS. However, considering the reviewers’ final assessments, I do not find that the paper’s broader claims are convincingly supported by sufficient evidence.

Specifically, Reviewer EAcX raised concerns about the theoretical analysis, including the independence assumption in the variance-reduction argument, the invalid implication from minimizing an MSE objective to a Lipschitz-type guarantee, and the assumption that semantic neighbor selection is reliable at the beginning of training. Although the authors revised parts of the theory, the resulting analysis still appears to rely on strong simplifying assumptions and serves more as intuition than as a rigorous justification of the proposed training procedure.

Reviewer 3HA2 also raised concerns about the strength and currency of the empirical evidence. The reviewer noted that the paper appears outdated and has no baselines after 2024, making it difficult to assess whether the method is competitive with recent developments in self-supervised visual representation learning. The reviewer also pointed out that the novelty is moderate and that the comparison to stronger recent baselines is not sufficiently convincing. The DINOv2 comparison was also acknowledged by the authors as originally unfair, and while the revised version improves the protocol labeling, the reviewer's final assessment indicates that the broader concern about controlled and up-to-date comparisons remains.

Overall, I find that the paper supports the specific claim that the proposed target construction can improve over I-JEPA in the reported settings. However, the evidence does not convincingly support the stronger claims of broadly superior representation learning, strong baseline competitiveness, or theoretically grounded improvement. Given the remaining concerns raised by the reviewers regarding theory and the lack of recent baselines, I cannot conclude that the claims are supported by sufficiently convincing evidence.

**Resubmission Of Major Revision:**

The authors may consider submitting a major revision at a later time.